# Regulation of sleep by cholinergic neurons located outside the central brain in *Drosophila*

**Joseph D. Jones, Brandon L. Holder, Kiran R. Eiken, Alex Vogt, Adriana I. Velarde, Alexandra J. Elder, Jennifer A. McEllin, Stephane Dissel** ⓘ *

Division of Biological and Biomedical Systems, School of Science and Engineering, University of Missouri-Kansas City, Kansas City, Missouri, United States of America

* dissels@umkc.edu

**Data Availability Statement:** All relevant data are within the paper and its Supporting Information files.

## Abstract

Sleep is a complex and plastic behavior regulated by multiple brain regions and influenced by numerous internal and external stimuli. Thus, to fully uncover the function(s) of sleep, cellular resolution of sleep-regulating neurons needs to be achieved. Doing so will help to unequivocally assign a role or function to a given neuron or group of neurons in sleep behavior. In the *Drosophila* brain, neurons projecting to the dorsal fan-shaped body (dFB) have emerged as a key sleep-regulating area. To dissect the contribution of individual dFB neurons to sleep, we undertook an intersectional Split-GAL4 genetic screen focusing on cells contained within the 23E10-GAL4 driver, the most widely used tool to manipulate dFB neurons. In this study, we demonstrate that 23E10-GAL4 expresses in neurons outside the dFB and in the fly equivalent of the spinal cord, the ventral nerve cord (VNC). Furthermore, we show that 2 VNC cholinergic neurons strongly contribute to the sleep-promoting capacity of the 23E10-GAL4 driver under baseline conditions. However, in contrast to other 23E10-GAL4 neurons, silencing these VNC cells does not block sleep homeostasis. Thus, our data demonstrate that the 23E10-GAL4 driver contains at least 2 different types of sleep-regulating neurons controlling distinct aspects of sleep behavior.

## Introduction

Sleep is a complex behavior that has been described in a variety of species ranging from jellyfish to humans [1]. Although the precise function of sleep remains unknown, current evidence indicates that sleep is required for maintaining optimal physiological and behavioral performance [2]. Over the last 20 years, multiple studies have demonstrated that the mechanisms and regulation of sleep are largely conserved from flies to mammals [3]. As in mammals, sleep-regulating centers are found in multiple areas of the *Drosophila* brain (Fig 1A) [3,4]. This compartmentalized organization of sleep-regulating regions in the brain probably underlies different functions and aspects of sleep behavior. Therefore, unequivocally assigning a role for a specific neuron or group of neurons in sleep regulation is a fundamental endeavor that will help uncover the function or functions of sleep.

The fan-shaped body (FB) is part of the central complex in the *Drosophila* brain, a region organized into multiple layers that plays a role in locomotion control [5], courtship behavior [6] and memory [7,8], nociception [9], visual feature recognition [10] and processing [11], social

**Funding:** This project was funded by a UMKC startup fund to SD. The funders had no role in study design, data collection and analysis, decision to publish, or preparation of the manuscript.

**Competing interests:** The authors have declared that no competing interests exist.

**Abbreviations:** AD, activation domain; dFB, dorsal fan-shaped body; DBD, DNA-binding domain; FB, fan-shaped body; GRN, gustatory receptor neuron; SEZ, subesophageal zone; SNAP, sleep-nullifying apparatus; TPN1, taste projection neurons 1; VLPO, ventrolateral preoptic nucleus; VNC, ventral nerve cord; VNC-SP, VNC sleep-promoting.

behaviors [12], and feeding decision-making [13]. In addition, dFB neurons have emerged as a key sleep-regulating area as their activation promotes sleep while their neuronal silencing reduces sleep [14–22]. Furthermore, sleep deprivation increases the excitability of dFB neurons, suggesting a role for the dFB in sleep homeostasis [17]. Finally, it was proposed that increasing sleep pressure switches dFB neurons from an electrically silent to an electrically active state. This process is regulated by dopaminergic signaling to the dFB [23] and the accumulation of mitochondrial reactive oxygen species in dFB neurons [24]. The physiological properties of dFB neurons have led to the suggestion that these cells are functionally analogous to the ventrolateral preoptic nucleus (VLPO), a key sleep-regulating center in the mammalian brain [17,25].

Recent studies have used 23E10-GAL4 as a driver to manipulate and monitor dFB neurons [17,19,21–24,26–31]. Because of its somewhat restricted expression pattern and strong capacity to modulate sleep, 23E10-GAL4 has become the prominent dFB driver. Analysis of an electron microscopy-based connectome of the central complex indicated that the 23E10-GAL4 driver expresses in 31 dFB neurons in the brain [32]. However, the extent to which these 31 dFB neurons function as a homogeneous group remains unclear.

In this study, we sought to identify the 23E10-GAL4 dFB neurons that are important for sleep regulation by conducting a targeted, intersectional Split-GAL4 [33] screen focused on 23E10-GAL4 dFB neurons. We report here that the 23E10-GAL4 driver expresses in many non-dFB neurons in the brain and in the ventral nerve cord (VNC), the fly equivalent to the spinal cord. Furthermore, we have identified 2 non-dFB 23E10-GAL4 expressing, sleep-regulating cholinergic neurons located in the VNC. Our analyses reveal that thermogenetic and optogenetic activation of these neurons promote sleep while silencing increases wakefulness. In addition, we provide evidence that these VNC sleep-promoting neurons are not involved in sleep homeostasis while other 23E10-GAL4 expressing cells are. Thus, we conclude that 23E10-GAL4 contains at least 2 different types of sleep-regulating populations, the 2 VNC neurons we are describing in this study and another group, likely made of dFB neurons, that regulate sleep homeostasis.

## Results

### The 23E10-GAL4 driver promotes sleep and expresses in many non-dFB neurons in the central nervous system

Previous studies have reported that increasing the activity of dFB neurons using the 23E10-GAL4 driver increases sleep [19–22]. To confirm these observations, we expressed the thermogenetic TrpA1 cation channel, which is activated by transferring flies to 31°C [34], in 23E10-GAL4 expressing neurons. Individual flies were loaded in *Drosophila* Activity Monitors (DAM2, Trikinetics) and assessed for 2 baseline days at 22°C, before raising the temperature to 31°C for 24 h on day 3 (Fig 1B). Sleep was defined as any period of inactivity lasting for at least 5 min, as previously described [35]. Consistent with previous reports, raising the temperature changes the sleep profile of both parental controls and results in a loss of sleep [36,37] (Fig 1C and 1D), whereas activating 23E10-GAL4 neurons strongly increases sleep (Fig 1E and 1F). Importantly, activation of 23E10-GAL4 neurons does not affect the amount of locomotor activity when the flies are awake (Fig 1G). Taken together, these data confirm that activating 23E10-GAL4 neurons promotes sleep and that the increase in sleep is not caused by a general motor defect. The 23E10-GAL4 driver is often chosen to manipulate dFB neurons and is widely considered as a "dFB-specific" tool [17, 19, 21–24, 26–31]. However, GAL4 lines commonly express in neurons outside the region of interest. To clarify the expression pattern of 23E10-GAL4, we expressed GFP under its control and identified more than 50 GFP-positive neurons in the brain, only half of which are dFB neurons (Fig 1H and 1I, and S1 Table). In addition, 23E10-GAL4 expresses in about 18 neurons in the VNC (See also S1 and S2 Movies). These results indicate that it is impossible to unequivocally

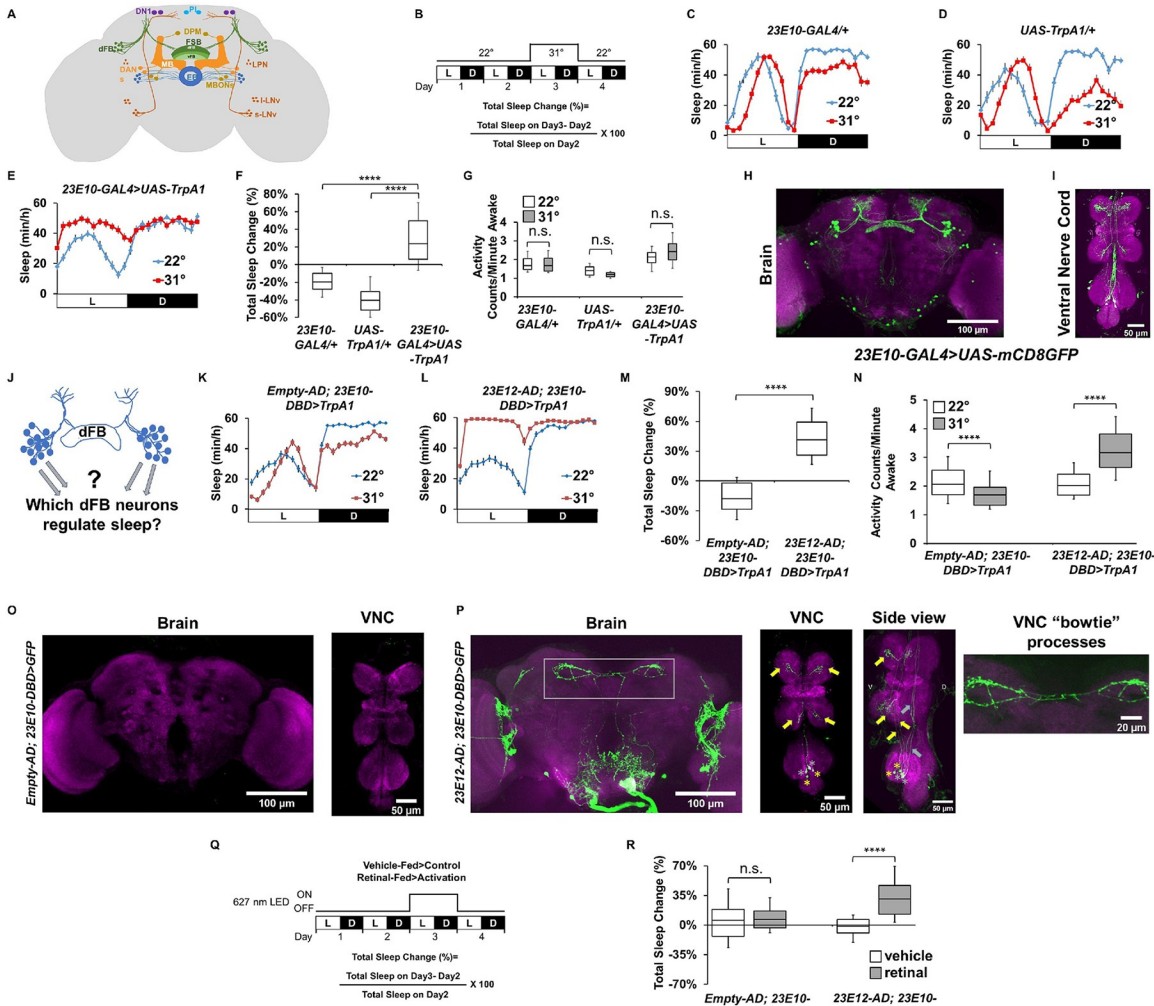

**Fig 1. The 23E10-GAL4 driver contains non-dFB sleep-promoting neurons.** (**A**) Cartoon depicting known sleep-regulating centers in the fly brain. (**B**) Diagram of the experimental assay. Sleep was measured at 22°C for 2 days to establish baseline sleep profile. Flies were shifted to 31°C for 24 h at the start of day 3 to increase activity of the targeted cells by activating the TrpA1 channel and then returned to 22°C. White bars (L) represent the 12 h of light and black bars (D) represent the 12 h of dark that oscillate daily. (**C, D**) Sleep profile in minutes of sleep per hour for day 2 (22°C, blue line) and day 3 (31°C, red line) for parental control female flies: *23E10-GAL4/+* (C) and *UAS-TrpA1/+* (D). (**E**) Sleep profile in minutes of sleep per hour for day 2 (22°C, blue line) and day 3 (31°C, red line) for *23E10-GAL4>UAS-TrpA1* female flies. (**F**) Box plots of total sleep change in % ((total sleep on day 3-total sleep on day 2/total sleep on day 2) × 100) for data presented in (C–E). Flies expressing UAS-TrpA1 in 23E10-GAL4 significantly increase sleep when switched to 31°C compared with parental controls, Kruskal–Wallis ANOVA followed by Dunn's multiple comparisons. ****$P < 0.0001$, $n = 25$–31 flies per genotype. (**G**) Box plots of locomotor activity counts per minute awake for flies presented in (C–E). Two-way repeated measures ANOVA followed by Sidak's multiple comparisons test found no differences in locomotor activity between 22°C and 31°C, n.s. = not significant, $n = 25$–31 flies per genotype. (**H, I**) Representative confocal stacks of a female *23E10-GAL4>UAS-mCD8GFP* brain (H) and VNC, (I). Green, anti-GFP; magenta, anti-nc82 (neuropile marker). (**J**) Cartoon depiction of the original goal of our Split-Gal4 screen, which was to identify which dFB neurons modulate sleep. (**K**) Sleep profile in minutes of sleep per hour for day 2 (22°C, blue line) and day 3 (31°C, red line) for empty control (*Empty-AD; 23E10-DBD>TrpA1*) flies. (**L**) Sleep profile in minutes of sleep per hour for day 2 (22°C, blue line) and day 3 (31°C, red line) for *23E12-AD; 23E10-DBD>TrpA1* flies. (**M**) Box plots of total sleep change in % for female control (Empty-AD; 23E10-DBD) and 23E12-AD; 23E10-DBD flies expressing UAS-TrpA1. A two-tailed unpaired *t* test revealed that *23E12-AD; 23E10-DBD>TrpA1* flies increase sleep significantly more than control flies when transferred to 31°C. ****$P < 0.0001$, $n = 44$–51 flies per genotype. (**N**) Box plots of locomotor activity counts per minute awake for flies presented in (M). Two-way repeated measures ANOVA followed by Sidak's multiple comparisons test found that locomotor activity per awake time is increased in *23E12-AD; 23E10-DBD>TrpA1* flies transferred to 31°C. ****$P < 0.0001$, $n = 44$–51 flies per genotype. (**O**) Representative confocal stacks of an *Empty-AD; 23E10-DBD>UAS-mCD8GFP* female brain (left panel) and VNC (right panel). Green, anti-GFP; magenta, anti-nc82 (neuropile marker). (**P**) Representative confocal stacks of an *23E12-AD; 23E10-DBD>UAS-mCD8GFP* female brain (left panel), VNC (middle panel left), a side view of the VNC (middle panel right) as well as a magnified view of VNC "bowtie" processes in the brain as indicated by the gray rectangle. Yellow arrows indicate TPN1-like processes in the VNC. Yellow asterisks indicate TPN1-like cell bodies. Gray arrows indicate "bowtie" neurons processes in the VNC. Gray asterisks indicate "bowtie" neurons cell bodies. Green, anti-

GFP; magenta, anti-nc82 (neuropile marker). (**Q**) Diagram of the experimental assay. Sleep was measured in retinal-fed and vehicle-fed flies for 2 days without 627 nm LED stimulation to establish baseline sleep profile. LEDs were then turned on for 24 h at the start of day 3 to increase activity of the targeted cells by activating the CsChrimson channel and then turned off on day 4. White bars (L) represent the 12 h of light and black bars (D) represent the 12 h of dark that are oscillating daily. (**R**) Box plots of total sleep change in % ((total sleep on day 3-total sleep on day 2/total sleep on day 2) × 100) for control (Empty-AD; 23E10-DBD) and 23E12-AD; 23E10-DBD female flies expressing CsChrimson upon 627 nm LED stimulation. Two-way ANOVA followed by Sidak's multiple comparisons revealed that total sleep is significantly increased in *23E12-AD; 23E10-DBD>UAS-CsChrimson* female flies stimulated with 627 nm LEDs when compared with vehicle-fed flies. ****$P < 0.0001$, n.s. = not significant, $n$ = 24–32 flies per genotype and condition. The raw data underlying parts F, G, M, N, and R can be found in S1 Data. DANs, dopaminergic neurons; dFB, dorsal fan-shaped body; DN1, dorsal neurons 1; DPM, dorsal paired medial neurons; EB, ellipsoid body; FSB, fan-shaped body; l-LNv, large lateral neurons ventral; LPN, lateral posterior neurons; MB, mushroom body; MBONs, mushroom body output neurons; PI, pars intercerebralis; s-LNv, small lateral neurons ventral; vFB, ventral fan-shaped body; VNC, ventral nerve cord.

assign a role in sleep promotion to 23E10-GAL4 dFB neurons (Fig 1J). Finally, since a recent study has implicated neurons located in the legs in sleep homeostasis [38], we also examined expression in the legs, gut, and ovaries in *23E10-GAL4>UAS-GFP* flies. Our analysis revealed no GFP staining in these structures (S1A–S1D Fig). While it is possible that 23E10-GAL4 expresses in other non-CNS cells that we have not assessed in this study, we feel that this is extremely unlikely. It is even more unlikely that these non-CNS neurons would regulate sleep. Thus, we conclude that the sleep-promoting properties of the 23E10-GAL4 driver likely originate in the CNS.

## The 23E10-GAL4 driver contains sleep-promoting neurons that are not dFB neurons

To shed light on the role of individual 23E10-GAL4 expressing neurons in sleep regulation, we employed a Split-GAL4 strategy [33] to access a reduced number of 23E10-GAL4 expressing cells. We focused primarily on the dFB in designing this screen (Fig 1J), a choice dictated by the number of studies supporting a role for this structure in sleep regulation [14–17,21,23,24,28,39]. Central to the Split-GAL4 technology is the fact that the functional GAL4 transcription factor can be separated into 2 non-functional fragments, a GAL4 DNA-binding domain (DBD) and an activation domain (AD). Different enhancers are used to express these 2 fragments and the functional GAL4 transcription factor is reconstituted only in the cells in which both enhancers are active [33]. In this screen, we created new Split-GAL4 drivers by combining AD and DBD lines that are putatively expressing in the dFB, as assessed by the expression pattern of their corresponding GAL4 lines. The complete description and results of this ongoing Split-GAL4 dFB-based screen are beyond the scope of this manuscript and will be described elsewhere.

However, when performing this screen, we identified a line (23E12-AD; 23E10-DBD) that strongly promotes sleep when thermogenetically activated with the TrpA1 channel (Fig 1L and 1M) when compared with an enhancer-less (empty) AD construct combined with 23E10-DBD (Fig 1K and 1M). Analysis of activity counts during awake time reveals that the increase in sleep observed when activating neurons contained in the 23E12-AD; 23E10-DBD line is not caused by a reduction in locomotor activity. On the contrary, these flies display an increase in waking activity upon neuronal activation (Fig 1N). These data demonstrate that activating 23E12-AD; 23E10-DBD neurons does not create a state of paralysis or motor defects as flies do not display any motor deficits when not sleeping. The increase in waking activity is likely explained by the need for a fly to perform tasks that are mutually exclusive to sleep in a reduced amount of waking time. Although increases in total sleep are indicative of increased sleep quantity, this measurement does not provide information about its quality or depth. To assess whether sleep quality is modulated by 23E12-AD; 23E10-DBD neurons, we analyzed sleep consolidation during the day and night in these flies and found that daytime and nighttime sleep bout duration is significantly increased in *23E12-AD; 23E10-DBD>UAS-TrpA1* flies upon neuronal activation (S2A and S2B Fig). Because increased sleep bout duration is believed

to be an indication of increased sleep depth [40], these data suggest that activating 23E12-AD; 23E10-DBD neurons not only increases sleep quantity, but also probably increases sleep depth. In addition, since sleep in *Drosophila* is sexually dimorphic [41–43], we also systematically assessed male flies in our experiments. As seen in S2C–S2F Fig, we obtained almost identical behavioral data when analyzing male flies using our thermogenetic activation approach.

Because the goal of our screen was to identify sleep-regulating dFB neurons in the 23E10-GAL4 pattern of expression, we expected that the 23E12-AD; 23E10-DBD Split-GAL4 line would express in dFB cells. Surprisingly, our anatomical analyses revealed that it does not express in any dFB neurons (Fig 1P and S3 and S4 Movies). Instead, this line expresses in 2 clusters of 4 to 5 neurons located in the anterior ventrolateral protocerebrum and in 4 VNC cells located in the metathoracic ganglion. A close examination of the anatomy of these 4 VNC neurons indicates that they have processes in the brain. Two of these neurons show a specific pattern of expression, with brain processes located in the superior medial protocerebrum and were named "bowtie" neurons (right panel in Fig 1P (see also S3 Movie)). An inspection of the expression pattern of 23E10-GAL4 (S1 Movie) confirms that these "bowtie" processes are present in this GAL4 driver, but they are difficult to observe because of their very close proximity to 23E10-GAL4 dFB projections. To further confirm that "bowtie" processes are unequivocally present in 23E10-GAL4, we have produced a confocal stack focused only on these structures in a *23E10-GAL4>UAS-mCD8GFP* brain (S1E Fig). 23E12-AD; 23E10-DBD additionally labels 2 neurons in the VNC whose somas are located very close to the "bowtie" neurons. These neurons have characteristic processes in each leg ganglion (yellow arrows in Fig 1P-VNC image). Based on their anatomy, we hypothesize that these cells could be taste projection neurons 1 (TPN1) that receive inputs from sweet gustatory receptor leg neurons (GRNs) and convey this information to the brain in the subesophageal zone (SEZ) [44]. Since all the neurons contained in the 23E12-AD; 23E10-DBD Split-GAL4 are part of the 23E10-GAL4 expression pattern, these data indicate that this driver contains non-dFB sleep-promoting cells.

## Optogenetic confirmation of sleep-promoting capacity of 23E12-AD; 23E10-DBD neurons

Because identifying non-dFB 23E10-GAL4 sleep-promoting neurons was unexpected, we sought to confirm our thermogenetic findings using an optogenetic approach with CsChrimson [45] (Fig 1Q). In this setup, optogenetic activation of 23E12-AD; 23E10-DBD neurons strongly increases total sleep, as well as daytime and nighttime sleep bout duration (Figs 1R, S3B and S3D). These effects are not seen in control or vehicle-fed *23E12-AD; 23E10-DBD>UAS-CsChrimson* flies (Figs 1R, S3A and S3C). Assessment of waking locomotor activity reveals that optogenetic activation does not affect waking activity in long sleeping retinal-fed females (S3E and S3F Fig), ruling out the possibility that locomotor defects are the underlying cause of the sleep phenotypes observed. Similar behavioral data were obtained with male flies (S4 Fig). Taken together, our anatomical and behavioral data clearly demonstrate that 23E10-GAL4 contains non-dFB sleep-promoting neurons.

## The 23E10-GAL4 driver contains sleep-promoting neurons that are in the VNC

To reveal the identity of the non-dFB 23E10-GAL4 sleep-promoting neurons, we undertook a second screen creating additional Split-GAL4 lines focusing on neurons that are present in the 23E12-AD; 23E10-DBD line using Color Depth MIP mask search [46]. We identified several AD lines that we combined with our 23E10-DBD stock. An example of such a line is shown in Fig 2A–2D. This Split-GAL4 line (VT020742-AD; 23E10-DBD) expresses only in 4 VNC

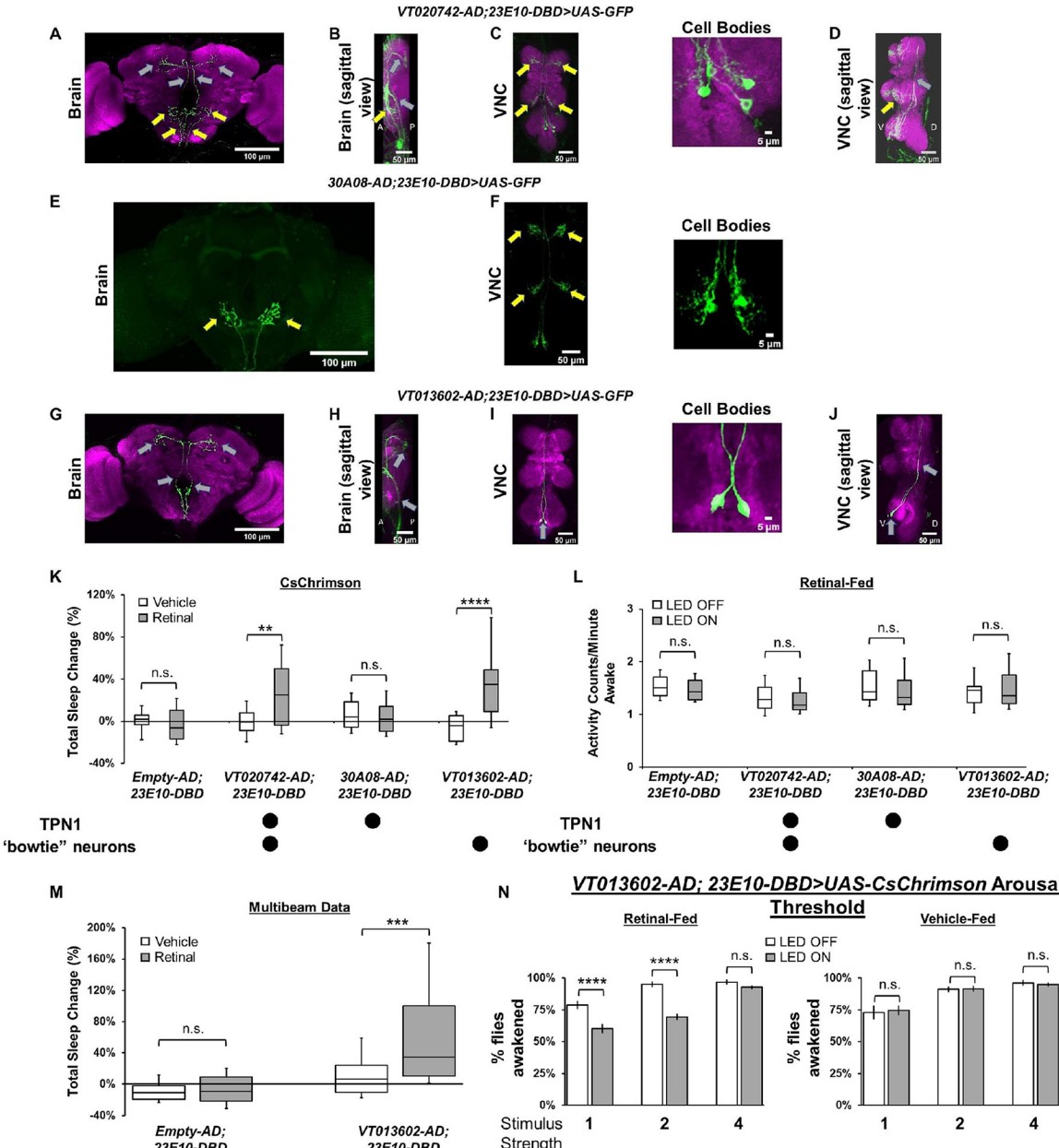

**Fig 2. The 23E10-GAL4 driver contains sleep-promoting neurons that are located in the VNC.** (**A-D**) Representative confocal stacks of a female *VT020742-AD; 23E10-DBD>UAS-GFP* brain (A), brain sagittal view (B), VNC (C), and VNC sagittal view (D). The cell bodies panel in (C) is a magnified view of the metathoracic area of the VNC. Yellow arrows indicate TPN1-like processes and gray arrows "bowtie" processes. Green, anti-GFP; magenta, anti-nc82 (neuropile marker). A = anterior, P = posterior, V = ventral, and D = dorsal. (**E, F**) Representative confocal stacks of a female *30A08-AD; 23E10-DBD>UAS-GFP* brain (E) and VNC (F). The cell bodies panel in (F) is a magnified view of the metathoracic area of the VNC. Yellow arrows indicate TPN1 processes in the brain and VNC. Green, anti-GFP. (**G–J**) Representative confocal stacks of a female *VT013602-AD; 23E10-DBD>UAS-GFP* brain (G), brain sagittal view (H), VNC (I), and VNC sagittal view (J). The cell bodies panel in (I) is a magnified view of the metathoracic area of the VNC. Gray arrows indicate "bowtie" processes and cell bodies in (I) and (J). Green, anti-GFP; magenta, anti-nc82 (neuropile marker). A = anterior, P = posterior, V = ventral, and D = dorsal. (**K**) Box plots of total sleep change in % for control (*Empty-AD; 23E10-DBD>UAS-CsChrimson*), *VT020742-AD; 23E10-DBD>UAS-CsChrimson*, *30A08-AD; 23E10-DBD>UAS-CsChrimson* and *VT013602-AD; 23E10-DBD>UAS-CsChrimson* vehicle-fed and retinal-fed female flies upon 627-nm LED stimulation. Two-way ANOVA followed by Sidak's multiple comparisons revealed that retinal-fed *VT020742-AD; 23E10-DBD>UAS-CsChrimson* and *VT013602-AD; 23E10-DBD>UAS-CsChrimson* flies increase sleep significantly when stimulated with 627-nm LEDs when compared with vehicle-fed flies. ** $P < 0.01$, **** $P < 0.0001$, n.s. = not significant, $n = 13–34$ flies per genotype and condition. (**L**) Box plots of locomotor activity counts per minute awake for retinal-fed flies presented in (K). Two-way repeated measures ANOVA followed by Sidak's multiple comparisons test show that locomotor activity per awake time is not affected when the flies are stimulated with 627-nm

LEDs, n.s. = not significant, n = 15–34 flies per genotype. (**M**) Box plots of total sleep change in % for control (*Empty-AD; 23E10-DBD>UAS-CsChrimson*) and *VT013602-AD; 23E10-DBD>UAS-CsChrimson* vehicle-fed and retinal-fed female flies upon 627-nm LED stimulation measured with the multibeam MB5 system. Two-way ANOVA followed by Sidak's multiple comparisons revealed that retinal-fed *VT013602-AD; 23E10-DBD>UAS-CsChrimson* flies increase sleep significantly when stimulated with 627-nm LEDs when compared with vehicle-fed flies. ***$P < 0.001$, n.s. = not significant, *n* = 19–24 flies per genotype and condition. (**N**) Arousal threshold in vehicle-fed and retinal-fed *VT013602-AD; 23E10-DBD>UAS-CsChrimson* female flies. Percentage of flies awakened by a stimulus of increasing strength (1, 2, and 4 downward movements in the SNAP apparatus) with and without 627-nm LEDs stimulation. Two-way ANOVA followed by Sidak's multiple comparisons indicates that in retinal-fed flies activation of VT013602-AD; 23E10-DBD neurons reduce the responsiveness to the 1 and 2 stimulus strength when compared with non-activated flies. No difference in responsiveness is seen at the strongest stimulus (4). Two-way ANOVA followed by Sidak's multiple comparisons indicates that in vehicle-fed flies, no difference in responsiveness is seen between LED stimulated and non-stimulated flies. ****$P < 0.0001$, n.s. = not significant, *n* = 16 flies per genotype and condition. The raw data underlying parts (K–N) can be found in S1 Data. SNAP, sleep-nullifying apparatus; TPN1, taste projection neurons 1; VNC, ventral nerve cord.

neurons (Fig 2C). Projections of 2 of these are "bowtie"-like (gray arrows in Fig 2A and 2B), while projections of the other 2 appear to be identical to TPN1 neurons that have been previously described (yellow arrows in Fig 2A–2D). VT020742-AD; 23E10-DBD does not express in any other neurons in the brain or VNC. To confirm the presence of TPN1 neurons in this Split-GAL4 line, we expressed GFP using a TPN1-specific driver, 30A08-LexA [44], while simultaneously expressing RFP using VT020742-AD; 23E10-DBD. GFP and RFP are co-expressed in 2 out of the 4 VT020742-AD; 23E10-DBD neurons (yellow arrows in S5A Fig). In addition, GFP and RFP signals are also found in the processes that penetrate each leg ganglion, but only RFP is expressed in the "bowtie" processes (gray arrows in S5A Fig). We thus conclude that VT020742-AD; 23E10-DBD labels 4 neurons in the metathoracic ganglion of the VNC: 2 "bowtie" neurons and 2 TPN1 cells. Optogenetic activation of these 4 neurons increases sleep in retinal-fed females (Fig 2K) and males (S6C Fig). In addition, daytime and nighttime sleep bout duration are significantly increased in retinal-fed *VT020742-AD; 23E10-DBD>UAS-CsChrimson* flies (S6A and S6B Fig for females and S6E and S6F Fig for males). Analysis of waking activity reveals that the sleep increases are not caused by a deficit in locomotor activity (Fig 2L for females and S6D Fig for males).

We next created a 30A08-AD; 23E10-DBD Split-GAL4 line to determine whether TPN1 neurons are the cells involved in sleep promotion in VT020742-AD; 23E10-DBD. This line labels only the 2 TPN1 neurons and no other cells in the brain or VNC (Fig 2E and 2F). Optogenetic activation of 30A08-AD; 23E10-DBD neurons has no effect on sleep (Figs 2K, S6A–S6F for males), indicating that activation of TPN1 neurons is not sufficient to induce sleep. To further rule out an involvement of TPN1 neurons in sleep promotion, we adopted the Split-GAL4 repressor Killer Zipper (KZip+) technology that inhibits an AD and DBD from forming a functional GAL4 [47]. Expressing KZip+ with 30A08-LexA while expressing GFP with VT020742-AD; 23E10-DBD reduces the number of GFP positive cells from 4 to 2 (S5B Fig), with only the 2 "bowtie" neurons observed in this combination. Thus, we can effectively subtract the TPN1 neurons from the VT020742-AD; 23E10-DBD pattern of expression. Optogenetic activation of the remaining "bowtie" neurons (in retinal-fed *VT020742-AD; 23E10-DBD>UAS-CsChrimson, 30A08-LexA>LexAop2-KZip+* flies) still promotes sleep (S5C Fig). Overall, these data indicate that the 2 "bowtie" neurons in VT020742-AD; 23E10-DBD increase sleep when activated and that TPN1 neurons are not involved in sleep promotion.

To further refine our analysis, we constructed an additional Split-GAL4 line (VT013602-AD; 23E10-DBD) that only expresses in the 2 "bowtie"-like VNC neurons (gray arrows in Fig 2G–2J and S5 and S6 Movies). In addition, we examined potential expression outside the CNS in *VT013602-AD; 23E10-DBD>UAS-GFP* flies and found no GFP expression in the legs, gut, or ovaries (S5D Fig). Because of the combinatorial nature of the Split-GAL4 system, necessitating an AD and DBD construct to be present in the same cell for a functional

GAL4 to be expressed, we feel that these data suggest that VT013602-AD; 23E10-DBD is likely only expressing in the 2 "bowtie"-like VNC neurons.

Optogenetic activation of retinal-fed VT013602-AD; 23E10-DBD neurons strongly promotes sleep (Figs 2K and S6C for male data) and increases daytime and nighttime bout duration (S6A, S6B, S6E and S6F Figs for males). Again, increases in sleep are not caused by abnormal locomotor activity (Figs 2L and S6D). Importantly, analysis of sleep parameters in vehicle-fed flies reveal that the sleep phenotypes observed are specific to activation of VT013602-AD; 23E10-DBD neurons (S7 and S8 Figs).

To further confirm that activation of VT013602-AD; 23E10-DBD neurons increases sleep, we employed the more sensitive multibeam activity monitors (MB5, Trikinetics) that contain 17 independent infrared beams, separated by 3 mm across the 51-mm tube length (S10A Fig). As seen in Fig 2M, optogenetic activation of retinal-fed VT013602-AD; 23E10-DBD neurons promotes sleep as assessed with the multibeam system. In addition, daytime sleep bout duration is increased in these flies while nighttime bout length is unchanged (S9 Fig). Importantly, these multibeam data confirm that VT013602-AD; 23E10-DBD neurons promote sleep when activated. In addition, we analyzed the position of the flies within the tube in the multibeam system (S10A Fig) and found that activating VT013602-AD; 23E10-DBD neurons does not modify the average position of a fly over the 24 h of optogenetic activation (S10B Fig). Thus, flies in which VT013602-AD; 23E10-DBD neurons have been activated do not localize closer to the food ruling out the possibility that these flies are increasing the duration of feeding (a behavior that could be mislabeled as sleep in our single beam DAM2 analysis). To further characterize the behavior of flies with activated VT013602-AD; 23E10-DBD neurons, we conducted additional experiments using video analysis. As seen in S10C Fig, we activated VT013602-AD; 23E10-DBD neurons for 1 h starting at ZT1 (1 h after lights turn on in the morning) and monitored the behaviors of individual flies. We found that time spent sleeping is significantly increased, while walking, grooming, and feeding durations are significantly decreased when VT013602-AD; 23E10-DBD neurons are activated (S10C Fig). Thus, we conclude that activating VT013602-AD; 23E10-DBD neurons increases both sleep and sleep consolidation but not another behavior that could be mislabeled as sleep (feeding or grooming) by our DAM2 analysis.

Since activating the 2 "bowtie" neurons increases daytime and nighttime sleep bout duration, we hypothesize that this manipulation increases sleep quality, and in particular, sleep depth. To investigate this possibility, we measured arousal thresholds in *VT013602-AD; 23E10-DBD>UAS-CsChrimson* flies. As seen in Fig 2N, the proportion of retinal-fed flies awakened by mild mechanical perturbations (perturbation levels 1 and 2) is significantly reduced in activated (LED ON) individuals compared with non-activated flies (LED OFF). Thus, these data demonstrate that when activated, the 2 "bowtie" neurons increase sleep quantity and sleep depth, as indicated by an increase in arousal threshold. Importantly, at the highest perturbation level (perturbation level 4), retinal-fed and LED-activated *VT013602-AD; 23E10-DBD>UAS-CsChrimson* flies respond to the stimulus similarly to controls, indicating that the sleep that is induced by activation of these neurons is reversible, an essential hallmark of sleep.

Taken together, our data demonstrate that we have identified 2 novel sleep-promoting neurons located in the VNC that are part of the 23E10-GAL4 expression pattern. We will from now on refer to these neurons as VNC-SP (VNC sleep-promoting).

Are VNC-SP neurons contained within the expression pattern of the 23E12-AD; 23E10-DBD and VT020742-AD; 23E10-DBD Split-GAL4 lines? While our anatomical analyses demonstrate that both these lines have "bowtie"-like processes within the brain (Figs 1P and 2A), it is possible that other VNC neurons have anatomical features similar to VNC-SP cells but are in fact different neurons. To assess whether the "bowtie" cells seen in

VT020742-AD; 23E10-DBD are VNC-SP neurons, we constructed a line that carries both VT020742-AD and VT013602-AD as well as 23E10-DBD. Expressing GFP with this line labels only 4 VNC neurons, 2 "bowtie" cells, and 2 TPN1 (S11A Fig, VNC panel). If the "bowtie" neurons were different in VT020742-AD; 23E10-DBD and VT013602-AD; 23E10-DBD, we would expect to see 6 cells. Since this is not the case, we can conclude that the 2 "bowtie" neurons observed in VT020742-AD; 23E10-DBD are VNC-SP neurons. To provide additional support to this finding, we created an additional Split-GAL4 line, VT013602-AD; VT020742-DBD. This Split-GAL4 line labels the 2 VNC-SP neurons with typical processes in the brain (gray arrows in S11B Fig) and promotes sleep when activated thermogenetically (S11C–S11E Fig).

To unmask the identity of the sleep-promoting neurons contained within the 23E12-AD; 23E10-DBD Split-GAL4 line, we conducted further experiments. We observed "bowtie"-like processes in the brain of *23E12-AD; VT020742-DBD>UAS-GFP* and *23E12-AD; VT013602-DBD>UAS-GFP* flies (gray arrows in S11F and S11G Fig). Thermogenetic activation of these 2 Split-GAL4 lines strongly increases sleep (S11H Fig). These data suggest that VNC-SP neurons are sleep-promoting cells that are contained within the expression pattern of the VT020742-AD; 23E10-DBD, 23E12-AD; 23E10-DBD, VT013602-AD; VT020742-DBD, 23E12-AD; VT020742-DBD, and 23E12-AD; VT013602-DBD Split-GAL4 lines.

## Silencing VNC-SP neurons reduces sleep

Having demonstrated that activation of VNC-SP neurons increases sleep duration and sleep depth, we wondered whether these cells could modulate sleep bi-directionally. To investigate this, we silenced the activity of VNC-SP neurons either chronically by expressing the inward rectifying potassium channel Kir2.1 [48] or acutely by using the temperature-sensitive Shi[ts1] construct [49]. Total sleep is reduced significantly when expressing Kir2.1 in VNC-SP neurons in female flies (Fig 3A and 3B). This reduction is accompanied by a decrease in the duration of both daytime (S12A Fig) and nighttime sleep bout (S12B Fig), suggesting a defect in sleep consolidation. Analysis of activity counts during waking time reveals that the decrease in sleep is not due to hyperactivity (Fig 3C). We sought to further validate our findings using the more sensitive multibeam system. As seen in Fig 3D, total sleep as assessed with the multibeam system is significantly decreased when expressing Kir2.1 in VNC-SP neurons. In addition, nighttime sleep bout duration is significantly decreased in these flies while daytime bout duration is unaffected (S12C and S12D Fig). We obtained identical behavioral results when expressing Kir2.1 in VNC-SP neurons of male flies (S12E–S12I Fig). Thus, chronic silencing of VNC-SP neurons decreases sleep and disrupts sleep consolidation.

Finally, we used an acute thermogenetic silencing protocol by expressing Shi[ts1] in VNC-SP neurons to confirm the findings obtained using chronic silencing. As expected, a shift in temperature reduces sleep in control flies (Fig 3E for females and S12J Fig for males). Acute silencing of VNC-SP neurons also reduces sleep, an effect particularly striking at night (Figs 3F, 3G, S12K and S12L for males). Importantly, nighttime sleep is significantly more reduced in *VNC-SP> Shi[ts1]* flies than in controls (Figs 3G and S12L).

Overall, our chronic and acute silencing data demonstrate that VNC-SP neurons positively regulate sleep. These results further reinforce the findings we obtained using optogenetic and thermogenetic activation experiments.

## VNC-SP neurons express and use acetylcholine to modulate sleep

Having demonstrated that VNC-SP neurons modulate sleep bi-directionally, we sought to identify the neurotransmitter(s) used by these cells. A previous study suggested that most

## Chronic silencing-Kir2.1

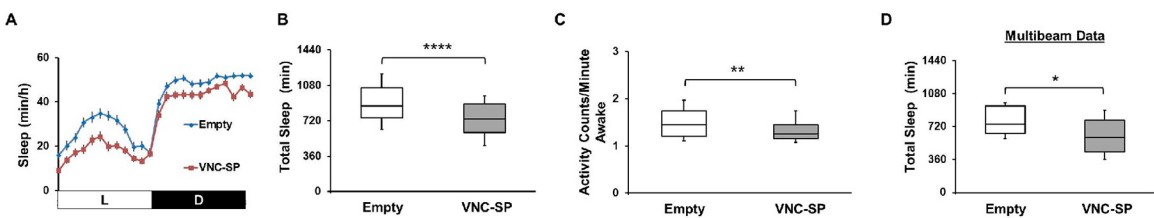

## Acute silencing-Shibire^ts1

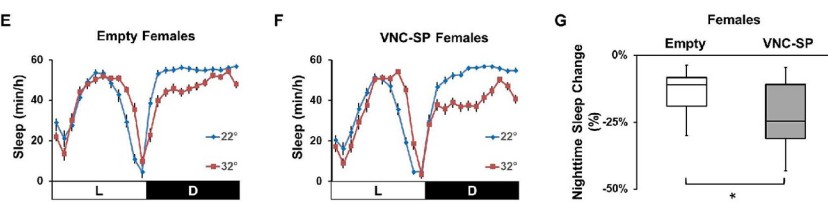

**Fig 3. Silencing VNC-SP neurons reduces sleep.** (**A**) Sleep profile in minutes of sleep per hour for control (*Empty-AD*; *23E10-DBD>UAS-Kir2.1*, blue line) and *VNC-SP>Kir2.1* (*VT013602-AD*; *23E10-DBD>UAS-Kir2.1*, red line) female flies. (**B**) Box plots of total sleep time (in minutes) for flies presented in (A). A two-tailed unpaired *t* test revealed that total sleep is significantly reduced in *VNC-SP>Kir2.1* female flies compared to controls. ****$P < 0.0001$, $n = 58$–60 flies per genotype. (**C**) Box plots of locomotor activity counts per minute awake for flies presented in (A). Two-tailed Mann–Whitney U tests revealed that activity per minute awake is significantly reduced in *VNC-SP>Kir2.1* female flies compared to controls. **$P < 0.001$, $n = 58$–60 flies per genotype. (**D**) Box plots of total sleep time (in minutes) for control (*Empty-AD; 23E10-DBD>UAS-Kir2.1*) and *VNC-SP>UAS-Kir2.1* female flies measured with the multibeam MB5 system. A two-tailed unpaired *t* test revealed that total sleep is significantly reduced in *VNC-SP>Kir2.1* female flies compared to controls, *$P < 0.05$, $n = 28$–31 flies per genotype. (**E**) Sleep profile in minutes of sleep per hour for control (*Empty-AD; 23E10-DBD>UAS-Shi^ts1*) female flies at 22˚C (blue line) and 32˚C (red line). (**F**) Sleep profile in minutes of sleep per hour for *VNC-SP>UAS-Shi^ts1* female flies at 22˚C (blue line) and 32˚C (red line). (**G**) Box plots of nighttime sleep change in % for female flies presented in (E) and (F). Two-tailed unpaired *t* tests revealed that *VNC-SP>Shi^ts1* flies lose significantly more sleep when transferred to 32˚C compared with controls. *$P < 0.05$, $n = 27$–31 flies per genotype. The raw data underlying parts (B), (C), (D), and (G) can be found in S1 Data. VNC-SP, VNC sleep-promoting.

neurons in the VNC use 1 of 3 neurotransmitters: GABA, glutamate, or acetylcholine [50]. We thus expressed GFP in VNC-SP neurons and stained them with antibodies against GABA, the vesicular glutamate transporter (VGlut), and choline acetyltransferase (ChAT), the enzyme necessary to produce acetylcholine. We observed unambiguous co-labeling of GFP and ChAT in VNC-SP neurons suggesting that they are cholinergic (Fig 4A) but did not detect any co-labeling between GFP and VGlut or GABA (Fig 4B and 4C). To confirm this finding, we employed a combinatorial approach expressing RFP in VNC-SP cells and GFP driven by a ChAT-LexA [51] construct inserted within the endogenous ChAT locus. This construct ensures that the expression of LexA is controlled by ChAT regulatory sequences and should reflect transcriptional activity at the endogenous ChAT locus. In these flies, we observe co-labeling of GFP and RFP in VNC-SP neurons (gray arrows in S13A Fig). In addition, we employed a similar approach using 23E10-GAL4 to drive expression of RFP and observed co-labeling of GFP and RFP in 2 neurons in the metathoracic ganglion of the VNC (gray arrows in S13B Fig). Interestingly, while conducting this experiment, we found that 2 to 3 dFB neurons per brain hemisphere are co-labeled by 23E10-GAL4 and the ChAT-LexA construct (S13B Fig, bottom panels, red arrows). Taken together, these data indicate that VNC-SP neurons and some 23E10-GAL4 dFB neurons are cholinergic.

Although our data indicate that VNC-SP neurons are cholinergic, it remains unclear whether cholinergic transmission is needed for the sleep-modulating role of VNC-SP neurons.

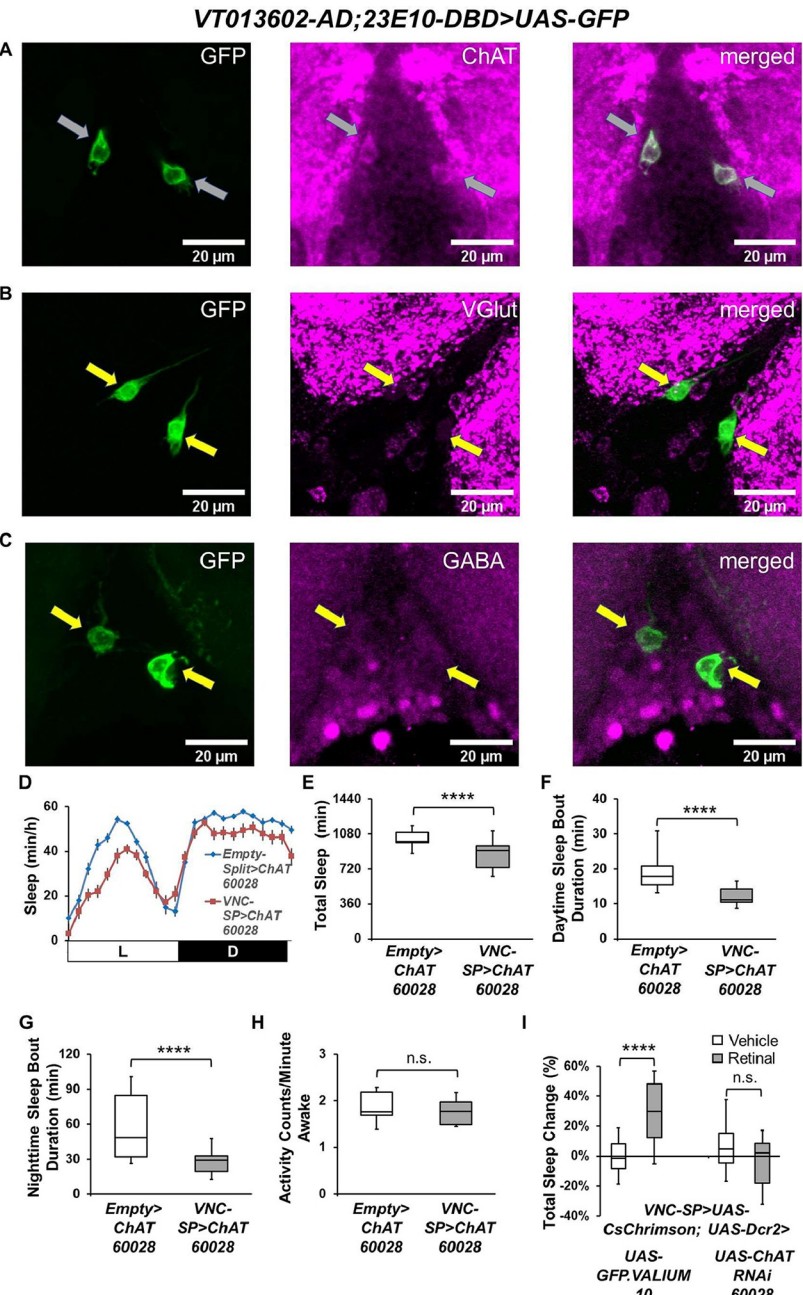

**Fig 4. VNC-SP neurons are cholinergic. (A–C)** Representative confocal stacks focusing on the metathoracic ganglion of the VNC of female *VT013602-AD; 23E10-DBD>UAS-GFP* flies stained with antibodies to ChAT (A), VGlut (B), and GABA (C). Gray arrows in (A) indicate colocalization of GFP and ChAT staining in VNC-SP neurons. Yellow arrows in (B) and (C) indicate the localization of VNC-SP neurons. Green, anti-GFP; magenta, anti-ChAT (A), anti-VGlut (B), anti-GABA (C). **(D)** Sleep profile in minutes of sleep per hour for control (*Empty-AD; 23E10-DBD>UAS-ChAT^RNAi 60028*, blue line) and *VNC-SP>ChAT^RNAi 60028* (*VT013602-AD; 23E10-DBD>UAS-ChAT^RNAi 60028*, red line) female flies. **(E)** Box plots of total sleep time (in minutes) for flies presented in (D). A two-tailed unpaired *t* test revealed that *VNC-SP> ChAT^RNAi 60028* flies sleep significantly less than controls. ****$P < 0.0001$, $n = 22$–30 flies per genotype. **(F)** Box plots of daytime sleep bout duration (in minutes) for flies presented in (D). A two-tailed Mann–Whitney U test revealed that *VNC-SP> ChAT^RNAi* flies daytime sleep bout duration is significantly reduced compared with controls. ****$P < 0.0001$, $n = 22$–30 flies per genotype. **(G)** Box plots of nighttime sleep bout duration (in minutes) for flies presented in (D). A two-tailed Mann–Whitney U test revealed that *VNC-SP> ChAT^RNAi* flies nighttime sleep bout duration is significantly reduced compared with controls. ****$P < 0.0001$, $n = 22$–30 flies per genotype. **(H)** Box plots of locomotor activity counts per minute awake for flies presented in (D). A two-tailed Mann–Whitney U test revealed that there is no difference in waking activity between

*VNC-SP> ChAT$^{RNAi}$* flies and controls, n.s. = not significant, *n* = 22–30 flies per genotype. (**I**) Box plots of total sleep change in % for vehicle-fed and retinal-fed *VT013602-AD; 23E10-DBD>UAS-CsChrimson; UAS-GFP.VALIUM10* (control) and *VT013602-AD; 23E10-DBD>UAS-CsChrimson; UAS-ChAT$^{RNAi\ 60028}$* flies stimulated by 627-nm LEDs. A two-way ANOVA followed by Sidak's multiple comparisons shows that sleep is significantly increased in control flies and that expressing ChAT RNAi in VNC-SP neurons completely abolishes the sleep-promoting effect of VNC-SP neurons activated by 627-nm LEDs. ****$P < 0.0001$, n.s. = not significant, *n* = 31–36 flies per genotype and condition. The raw data underlying parts (E–I) can be found in S1 Data. VNC, ventral nerve cord; VNC-SP, VNC sleep-promoting.

To address this question, we disrupted cholinergic transmission in VNC-SP neurons using 2 independent RNAi lines for ChAT. Out of these 2 lines, only 1 (60028) led to a significant reduction in ChAT levels, as judged by qPCR and immunocytochemistry, and was primarily used in our behavioral studies (S13C–S13E Fig). Behaviorally, reducing ChAT levels in VNC-SP neurons significantly reduces sleep (Fig 4D and 4E) as well as daytime and nighttime sleep bout duration in female flies (Fig 4F and 4G). The reduction in sleep is not due to hyper-activity (Fig 4H). We obtained similar behavioral data when examining males (S13F–S13I Fig). The second ChAT RNAi line (25856) only modestly reduces total sleep in female flies, when expressed in VNC-SP neurons. However, these flies show a significant increase in waking activity, suggesting that the sleep reduction may be caused by hyperactivity. No other sleep parameters are changed when expressing this weakest RNAi line in VNC-SP neurons in males or females (S13J–S13Q Fig). These behavioral data are in agreement with our quantification of RNAi efficiency (S13C–S13E Fig), with only the stronger RNAi line (60028) capable of significantly modulating sleep when expressed in VNC-SP neurons. Taken together, our data indicate that cholinergic transmission is involved in regulating the sleep-modulating effect of VNC-SP neurons. We also note that the behavioral effects of reducing ChAT levels and silencing VNC-SP neurons are consistent with each other, suggesting that in the absence of ChAT, the ability of VNC-SP neurons to regulate sleep is seriously diminished. To further determine whether cholinergic transmission is necessary for VNC-SP neurons to promote sleep, we activated VNC-SP neurons optogenetically, while simultaneously expressing ChAT RNAi in these cells. As seen in Fig 4I, expression of CsChrimson and an RNAi line targeting GFP (control RNAi) does not prevent sleep increase in retinal-fed flies. However, expressing ChAT RNAi completely abolishes the sleep promotion triggered by activation of VNC-SP neurons. In conclusion, our data demonstrate that VNC-SP neurons are cholinergic and that disruption of cholinergic transmission abolishes their sleep-promoting role.

To confirm that the sleep phenotypes we observe are due to VNC-SP neurons, we optogenetically activated VNC-SP cells using the VT013602-AD; 23E10-DBD Split-GAL4 line while expressing the Split-GAL4 repressor KZip+ [47] under the control of a ChAT-LexA driver. In this case, the presence of the KZip+ repressor in ChAT expressing neurons blocks the sleep induction caused by activation of VNC-SP neurons (S13R Fig) as well as the expression of GFP by VNC-SP neurons (S13S Fig). Taken together, these data confirm that the sleep pheno-types we observe are due to activation of VNC-SP neurons and further demonstrate that these cells are cholinergic.

Having identified the neurochemical identity of VNC-SP neurons, we further investigated the 23E12-AD; 23E10-DBD and VT020742-AD; 23E10-DBD Split-GAL4 lines. First, we found that the 2 "bowtie"-like VNC neurons contained in the 23E12-AD; 23E10-DBD line are expressing ChAT (gray arrows in S14A Fig, VNC panels). Expressing the Split-GAL4 repressor KZip+ under the control of a ChAT-LexA driver removes expression from these 2 "bowtie"-like cells (S14A and S14B Fig) and effectively blocks the sleep that is induced by thermogenetic activation of 23E12-AD; 23E10-DBD neurons (S14E Fig). These data indicate that VNC-SP

neurons are sleep-promoting cells within the 23E12-AD; 23E10-DBD expression pattern. Unsurprisingly, given our finding that VNC-SP neurons are contained in the VT020742-AD; 23E10-DBD Split-GAL4 line (S11A Fig), removing expression in VNC-SP neurons by expression of the KZip+ repressor under the control of a ChAT-LexA driver (S14C and S14D Fig) completely abolishes the increase in sleep induced by activation of VT020742-AD; 23E10-DBD neurons (S14E Fig). In conclusion, VNC-SP neurons are sleep-promoting neurons contained within the expression pattern of the 23E12-AD; 23E10-DBD and VT020742-AD; 23E10-DBD Split-GAL4 lines. Because we demonstrated that TPN1 cells do not regulate sleep (Fig 2K), VNC-SP neurons are the only sleep-promoting neurons in VT020742-AD; 23E10-DBD. For 23E12-AD; 23E10-DBD, we cannot rule out that other ChAT expressing neurons also regulate sleep in addition to VNC-SP cells.

Since the VNC receives and integrates sensory inputs from the periphery and sends this information to the brain [52–54], we wondered whether VNC-SP neurons could similarly send sleep-relevant signals to the brain. The location of VNC-SP neurons in the VNC and their processes in the brain suggests that this is a likely possibility. We thus expressed the dendritic marker DenMark [55] in conjunction with the presynaptic marker synaptotagmin-GFP (syt.eGFP) [56] in VNC-SP neurons. Our data indicate that VNC-SP neurons have mostly postsynaptic sites in the VNC, as expected if these cells receive sensory information from the periphery (S15 Fig, middle panels). However, and surprisingly, the brain processes of these cells are labeled with both the presynaptic and postsynaptic markers (S15 Fig, top panels). This suggests that VNC-SP neurons are well positioned to receive sensory inputs from the periphery in the VNC and that they are sending this information to the brain. However, since the brain processes of VNC-SP neurons also harbor postsynaptic sites, it is likely that they also receive inputs in the brain. Nevertheless, our data suggest that VNC-SP neurons send sleep-relevant signals to the brain.

## Does 23E10-GAL4 contain multiple sleep-regulating regions with functional specificity?

Our combined data demonstrate that VNC-SP neurons, which are part of the 23E10-GAL4 pattern of expression, promote sleep. Since 23E10-GAL4 is widely seen as dFB-specific driver, the identification of VNC-SP neurons is a major finding that may impact our current views about the role of the dFB in sleep regulation. Does our data mean that the dFB plays no role in sleep regulation? Based on the number of studies supporting a role for this structure in sleep and particularly in sleep homeostasis [14–17,21,23,24,28,39], we think that this is unlikely.

With this in mind, we wanted to assess whether silencing all 23E10-GAL4 neurons or just 23E10-GAL4 VNC-SP neurons separately would affect sleep homeostasis. Chronic silencing of all 23E10-GAL4 neurons by expression of Kir2.1 effectively blocks homeostatic sleep rebound when compared with both parental controls (Fig 5A). Interestingly, chronic silencing of VNC-SP cells does not disrupt homeostatic sleep rebound (Fig 5B). These data support a role for 23E10-GAL4 neurons in sleep homeostasis, in agreement with previous studies. However, this homeostatic role is not performed by VNC-SP neurons, indicating that these cells are involved in a different aspect of sleep behavior (Fig 5C).

## Discussion

Despite years of extensive research, the precise function or functions of sleep remain largely unknown. As in mammals, sleep is a behavior that is regulated by multiple areas of the *Drosophila* brain [3,4]. One of the goals of neuroscience is to ascribe behaviors to specific cell types. Thus, to fully uncover the function(s) of sleep, cellular resolution of sleep-regulating

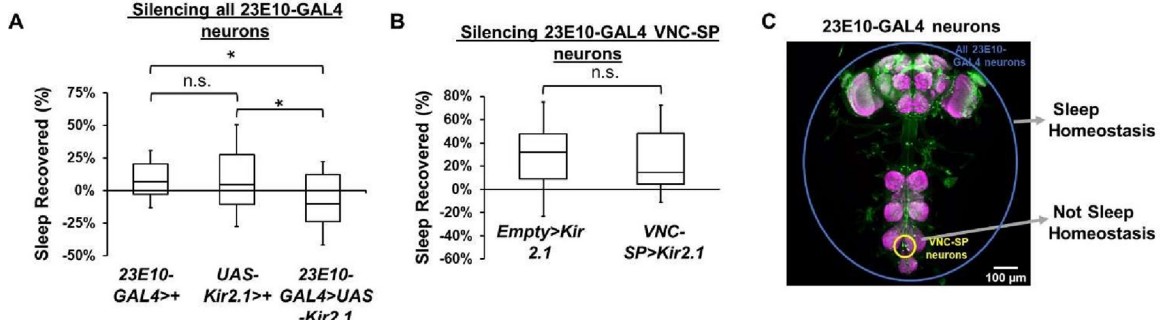

**Fig 5. Multiple sleep-regulating regions in 23E10-GAL4 with functional specificity?** (**A**) Box plots of total sleep recovered in % during the first 24 h following 12 h of sleep deprivation at night for *23E10-GAL4/+* and *UAS-Kir2.1/+* parental controls and *23E10-GAL4>UAS-Kir2.1* female flies. A one-way ANOVA followed by Tukey's multiple comparisons demonstrate that both parental controls have a bigger homeostatic sleep rebound than *23E10-GAL4>UAS-Kir2.1* female flies. *$P < 0.05$, n.s. = not significant, $n = 47$–62 flies per genotype. (**B**) Box plots of total sleep recovered in % during the first 24 h following 12 h of sleep deprivation at night for *Empty-AD; 23E10-DBD>UAS-Kir2.1* and *VT013602-AD; 23E10-DBD>UAS-Kir2.1* female flies. A two-tailed unpaired *t* test revealed that there is no difference in homeostatic sleep rebound, n.s. = not significant, $n = 24$–29 flies per genotype. (**C**) Representative confocal stacks of a female *23E10-GAL4>UAS-GFP* fly highlighting different 23E10-GAL4 expressing neurons and their different roles in sleep modulation. The raw data underlying parts (A) and (B) can be found in S1 Data. VNC-SP, VNC sleep-promoting.

neurons needs to be achieved. Doing so will help to unequivocally assign a role or function to a given neuron or group of neurons in sleep behavior.

With this in mind, we undertook an analysis of the neurons involved in sleep regulation in *Drosophila*, starting with the dFB/23E10-GAL4 expressing cells, a choice dictated by multiple studies supporting their role in sleep regulation [14–17,21,23,24,28,39]. The most widely used tool to manipulate dFB neurons is the 23E10-GAL4 driver [23,26], and thermogenetic or opto-genetic activation of 23E10-GAL4 neurons results in increased sleep [19–22]. However, whether 23E10-GAL4 dFB neurons can be considered as a functionally homogeneous group was unclear. Also unclear was whether the sleep promotion caused by activation of 23E10-GAL4 neurons is due to dFB neurons or to other cells within the 23E10-GAL4 pattern of expression.

In this study, we demonstrate that 23E10-GAL4 expresses in many neurons outside the dFB in the brain and in the VNC. Thus, to identify individual 23E10-GAL4 sleep-promoting neu-rons, we employed a Split-GAL4 strategy [33] and we demonstrate that 2 neurons located in the VNC (named VNC-SP neurons) modulate sleep. To our knowledge, our work provides the first unequivocal description of individual sleep-promoting 23E10-GAL4 expressing neu-rons. Activating these cells increases sleep, sleep consolidation, and sleep depth by increasing arousal thresholds. Moreover, chronic and acute silencing of VNC-SP neurons reduce sleep and sleep consolidation, confirming that VNC-SP neurons positively regulate sleep. Further-more, our study reveals that VNC-SP neurons are cholinergic and that reducing cholinergic transmission in VNC-SP neurons reduces sleep and sleep consolidation, an effect mirroring the silencing experiments. Finally, we demonstrate that VNC-SP neurons require cholinergic transmission to promote sleep when activated.

Where in the nervous system are these VNC-SP cholinergic signals received? We show that VNC-SP neurons receive synaptic inputs in the VNC and have presynaptic sites in the brain. Since VNC neurons are known to receive and integrate sensory inputs from the periphery and send this information to the brain [52–54], our data suggests that VNC-SP neurons are well positioned to collect sleep-relevant sensory inputs from the periphery and relay this information to the brain. The nature of these inputs will require additional investigations that will further uncover the function and role of VNC-SP neurons in sleep regulation. Known modulators of

sleep include the immune system and metabolic functions. Based on their location, it is possible that VNC-SP neurons integrate signals relevant to these 2 sleep-modulating systems.

While our study clearly demonstrates that VNC-SP neurons regulate sleep, it also raises some significant questions. First, VNC-SP neurons are not part of the dFB. Since 23E10-GAL4 has been widely used as a dFB-specific driver to manipulate and monitor dFB neurons, we believe that the identification of VNC-SP neurons raises some questions about the role of the dFB in sleep regulation and the use of 23E10-GAL4 as a dFB-specific tool. Existing data obtained with the 23E10-GAL4 driver may need to be reinterpreted in the light of our identification of VNC-SP neurons. Are the many phenotypes reported when using 23E10-GAL4 as a driver due to the dFB or to VNC-SP neurons? We believe that these important questions must be addressed in order to gain a more complete understanding of sleep regulation. Independently of the outcome of these inquiries, our data clearly highlight the need to obtain tools that are as specific as possible when attempting to link a given behavior with a specific neuron or group of neurons.

In an attempt to reconcile our findings with previous studies supporting a role for dFB neurons in sleep homeostasis, we show that 23E10-GAL4 expressing neurons are involved in sleep homeostasis, as silencing them completely abrogates sleep rebound following a night of sleep deprivation. However, these data do not rule out a role for non-dFB 23E10-GAL4 neurons in sleep homeostasis, though we believe this to be unlikely. Interestingly, we demonstrate that silencing VNC-SP neurons does not reduce homeostatic sleep rebound. Taken together, our data indicate that the 23E10-GAL4 driver contains at least 2 different types of sleep-regulating neurons that may underlie different roles or functions in sleep behavior. Some 23E10-GAL4 neurons (most likely dFB neurons) regulate sleep homeostasis while VNC-SP neurons regulate a different aspect of sleep behavior. Interestingly, a recent study proposed that the homeostatic sleep drive engages the dFB while a learning-dependent sleep drive relies on the activity of the ventral fan-shaped body [22]. Thus, it is likely that dFB and VNC-SP neurons are involved in different aspects and functions of sleep. Future studies will further investigate the role of VNC-SP neurons in sleep regulation.

In conclusion, our work identified a novel group of cholinergic sleep-promoting neurons located in the VNC in *Drosophila*. We show that these cells regulate sleep quantity and sleep depth and that they are perfectly positioned to receive and integrate sensory inputs from the periphery and send this information to the brain.

## Materials and methods

### *Drosophila* stocks and rearing

Flies were cultured at 25°C with 50% humidity under a 12-h light: 12-h dark cycle. Flies were kept on a standard yeast and molasses diet (per 1 L: 50 g yeast, 15 g sucrose, 28 g corn syrup, 33.3 mL molasses, 9 g agar). Fly stocks used in this study are listed in S2 Table.

### Sleep behavioral experiments

Sleep was assessed as previously described [18,35]. Approximately 4- to 10-day-old virgin females or males were used as described in figure legends. Briefly, flies were placed into individual 65-mm tubes and all activity was continuously measured through the DAM2 Trikinetics Drosophila Activity Monitoring System (www.Trikinetics.com; Waltham, Massachusetts). Locomotor activity was measured in 1-min bins and sleep was defined as periods of quiescence lasting at least 5 min. For multibeam monitoring, we used the MB5 monitors (www. Trikinetics.com, Waltham, Massachusetts).

## Quantitative video analysis

Video recording was performed on flies that were optogenetically activated. Flies fed food supplemented with 400 μM all-trans retinal (Sigma, #R2500) or vehicle were loaded into individual 65-mm tubes and placed on custom-built platforms. One webcam per platform was used to record the flies for 1 h starting at ZT1 (1 h after lights turn on in the morning). Fly behavior was manually scored and categorized as walking, grooming, feeding, resting (inactivity <5 min), or sleeping (inactivity >5 min). Videos were recorded on consecutive days for baseline (LEDs off) and activation (LEDs on). Three separate experiments were performed and the pooled data is shown.

## Optogenetic activation

For optogenetic activation, flies expressing CsChrimson were loaded into individual 65-mm tubes with food supplemented with 400 μM all-trans retinal (Sigma, #R2500) and kept in the dark for 48 h. Sleep was then monitored under low-light conditions for 48 h of baseline measurements. To activate CsChrimson, flies were put under 627-nm LEDs (LuxeonStar LXM2-PD01-0040) set to a pulse cycle of (5 ms on, 95 ms off) × 20 with a 4 s delay between pulse cycles. The light intensity of this protocol is 0.024 mW/mm$^2$.

## Sleep deprivation assay

Sleep deprivation was performed as previously described [18,57]. Briefly, flies were placed into individual 65-mm tubes in DAM2 monitors and the sleep-nullifying apparatus (SNAP) was used to sleep deprive flies for 12 h during the dark phase. Sleep homeostasis was calculated for each individual fly as the ratio of the minutes of sleep gained above baseline during 24 h of recovery sleep divided by the total min of sleep lost during 12 h of sleep deprivation.

## ChAT RNAi lines validation

We first sought to validate ChAT RNAi lines by crossing them to the pan-neuronal elav-GAL4 driver. One of the lines (60028) led to lethality when crossed to elav-GAL4, indicative of a significative reduction in ChAT levels. The second line (25856) gave rise to progeny suggesting lack of efficiency in reducing ChAT levels. Note that a similar approach was used to validate ChAT RNAi constructs in other studies [30]. However, to further confirm the efficiency of these ChAT RNAi constructs, we performed a qPCR analysis on flies expressing ChAT RNAi (60028 and 25856) specifically in the eyes using a GMR-GAL4 driver. In these experiments, we compared ChAT expression levels in the eyes and in the bodies of the flies. These analyses confirmed that only 1 line (60028) is efficient at reducing ChAT expression levels.

## Arousal threshold

Retinal or vehicle-fed VNC-SP>UAS-CsChrimson flies were placed into individual 65-mm tubes in Trikinetics monitors. These were then loaded in the SNAP sleep deprivation apparatus. The SNAP tilts asymmetrically from −60˚ to +60˚, such that sleeping flies are disturbed during each downward movement [58]. Each individual SNAP movement takes 15 s to complete. To assess arousal threshold, we modified the SNAP parameters so that flies were subjected to only 1, 2, or 4 downward movements in the SNAP every hour for 24 h and their locomotor responses were analyzed. We then averaged each individual hour response to obtain the percentage of flies awakened by a given strength of perturbation.

## Quantitative real-time PCR

QPCR were performed as previously described [18,59]. Briefly, total RNA was isolated from the bodies and eyes of approximately 10 flies per replicate with Trizol (Invitrogen, Carlsbad, California) and DNAse I digested. cDNA synthesis was performed using Superscript IV VILO (Invitrogen) according to manufacturer protocol. cDNA samples were loaded in duplicate and quantitatively analyzed using SYBR Green Master Mix (Applied Biosystems) and gene-specific primers using the Applied Biosystems 7500 system. Expression values for RP49 were used to normalize results between groups. The following primer sets were used: rp49: fw aagaagcgcaccaagcacttcatc, rev tctgttgtcgatacccttgggctt and ChAT: fw caccgagcgatacaggatgg, rev ggcaccttgggtagagtgtc.

## Immunocytochemistry

Flies were cold anesthetized, brains and VNC were dissected in ice cold Schneider's *Drosophila* Medium (Gibco, 21720024) and blocked in 5% normal goat serum. Following blocking flies were incubated overnight at 4°C in primary antibody, washed in PBST, and incubated overnight at 4°C in secondary antibody. Primary antibodies used were chicken anti-GFP (1:1,000; Aves Labs, #GFP-1020), mouse anti-bruchpilot (1:50; Developmental Studies Hybridoma Bank (DSHB), nc82-s), mouse anti-V5 (1:400; Sigma Aldrich, #V8012), rabbit anti-DsRed (1:250; Takara, #632496), mouse anti-ChAT (1:100; DSHB, Chat4b1-c), rabbit anti-VGlut (1:500; DiAntonio lab [60], Washington University), and rabbit anti-GABA (1:1,000; Sigma-Aldrich, #A2052). Secondary antibodies used were goat anti-chicken AlexFluor 488 (1:800; Invitrogen, #A32931), goat anti-mouse AlexaFluor 555 (1:400; Invitrogen #A32727), goat anti-rabbit AlexaFluor 555 (1:400; Invitrogen, #A32732), goat anti-mouse AlexaFlour 633 (1:400; Invitrogen, $A21052), and goat anti-rabbit AlexaFluor 633 (1:400; Invitrogen, #A21071). Brains and VNCs were mounted on polylysine-treated slides in Vectashield H-1000 mounting medium. Imaging was performed on a Zeiss 510 meta confocal microscope using a Plan-Apochromat 20× or Plan-Neofluar 40× objective. Z-series images were acquired with a 1 μM slice size using the same settings (laser power, gain, offset) to allow for comparison across genotypes. Images were processed and analyzed using ImageJ.

## Statistical analysis

Statistical analyses were performed with Prism9 software (GraphPad). Normal distribution was assessed with the D'Agostino–Pearson test. Normally distributed data were analyzed with parametric statistics: $t$ test, one-way analysis of variance or two-way ANOVA followed by the planned pairwise multiple comparisons as described in the legends. For data that significantly differed from the normal distribution, nonparametric statistics were applied, Mann–Whitney U test or Kruskal–Wallis test followed by Dunn's multiple test. Some non-normally distributed data were subjected to log transformation or Box–Cox transformation before two-way ANOVA followed by planned pairwise multiple comparisons as described in the legends. When box plots are used, the bottom and top of each box represents the first and third quartile, and the horizontal line dividing the box is the median. The whiskers represent the 10th and 90th percentiles. All statistically different groups are defined as $^*P < 0.05$, $^{**}P < 0.01$, $^{***}P < 0.001$, and $^{****}P < 0.0001$.

## Supporting information

**S1 Fig. Expression pattern of 23E10-GAL4. (A–D)** Representative confocal stack images of adult tissues from *23E10-GAL4 > UAS-mCD8GFP* female flies. GFP is expressed in the brain

(A), but not the gut (B), leg (C), or ovaries (D). Tissue was dissected, fixed, and stained with DAPI. Green, GFP; blue, DAPI. (**E**) Representative confocal stacks of a female *23E10-GAL4>UAS-mCD8GFP* brain centered on the "bowtie" processes from the VNC-SP neurons. Gray arrows indicate the processes of VNC-SP cells coming from the VNC. Area contained in the gray box is highlighted on the right. Green, anti-GFP; magenta, anti-nc82 (neuropile marker).
(TIF)

**S2 Fig. Thermogenetic activation of 23E12-AD; 23E10-DBD neurons. Additional data.** (**A**) Box plots of daytime sleep bout duration (in minutes) for female flies presented in Fig 1M. Two-way repeated measures ANOVA followed by Sidak's multiple comparisons test found that daytime sleep bout duration is significantly increased in *23E12-AD; 23E10-DBD>TrpA1* flies transferred to 31˚C. ****$P < 0.0001$, n.s. = not significant, $n = 44$–51 flies per genotype. (**B**) Box plots of nighttime sleep bout duration (in minutes) for female flies presented in Fig 1M. Two-way repeated measures ANOVA followed by Sidak's multiple comparisons test found that nighttime sleep bout duration is significantly increased in *23E12-AD; 23E10-DBD>TrpA1* flies transferred to 31˚C. **$P < 0.01$, ****$P < 0.0001$, $n = 44$–51 flies per genotype. (**C**) Box plots of total sleep change in % ((total sleep on day 3-total sleep on day 2/ total sleep on day 2) × 100) for male control (Empty-AD; 23E10-DBD) and 23E12-AD; 23E10-DBD flies expressing UAS-TrpA1. A two-tailed Mann–Whitney U test revealed that *23E12-AD; 23E10-DBD>TrpA1* flies increase sleep significantly more than control flies when transferred to 31˚C. ****$P < 0.0001$, $n = 43$–46 flies per genotype. (**D**) Box plots of daytime sleep bout duration (in minutes) for flies presented in (C). Two-way repeated measures ANOVA followed by Sidak's multiple comparisons test found that daytime sleep bout duration is significantly increased in *23E12-AD; 23E10-DBD>TrpA1* male flies transferred to 31˚C. *$P > 0.05$, ****$P < 0.0001$, $n = 43$–46 flies per genotype. (**E**) Box plots of nighttime sleep bout duration (in minutes) for flies presented in (C). Two-way repeated measures ANOVA followed by Sidak's multiple comparisons test found that nighttime sleep bout duration is not reduced in *23E12-AD; 23E10-DBD>TrpA1* flies transferred to 31˚C, contrary to controls. ****$P < 0.0001$, n.s. = not significant, $n = 43$–46 flies per genotype. (**F**) Box plots of locomotor activity counts per minute awake for flies presented in (C). Two-way repeated measures ANOVA followed by Sidak's multiple comparisons test found that locomotor activity per awake time is increased in *23E12-AD; 23E10-DBD>TrpA1* male flies transferred to 31˚C. **$P < 0.01$, ****$P < 0.0001$, $n = 43$–46 flies per genotype. The raw data underlying parts (A–F) can be found in S1 Data.
(TIF)

**S3 Fig. Optogenetic activation of 23E12-AD; 23E10-DBD neurons-female data.** (**A**) Box plots of daytime sleep bout duration (in minutes) for vehicle-fed flies presented in Fig 1R. Two-way repeated measures ANOVA followed by Sidak's multiple comparisons. ***$P < 0.001$, n.s. = not significant, $n = 25$–28 flies per genotype. (**B**) Box plots of daytime sleep bout duration (in minutes) for retinal-fed flies presented in Fig 1R. Two-way repeated measures ANOVA followed by Sidak's multiple comparisons revealed that daytime sleep bout duration is significantly increased in activated *23E12-AD; 23E10-DBD>UAS-CsChrimson* female flies. ****$P < 0.0001$, n.s. = not significant, $n = 24$–32 flies per genotype. (**C**) Box plots of nighttime sleep bout duration (in minutes) for vehicle-fed flies presented in Fig 1R. Two-way repeated measures ANOVA followed by Sidak's multiple comparisons, n.s. = not significant, $n = 25$–28 flies per genotype. (**D**) Box plots of nighttime sleep bout duration (in minutes) for retinal-fed flies presented in Fig 1R. Two-way repeated measures ANOVA followed by Sidak's multiple comparisons revealed that nighttime sleep bout duration is significantly increased in activated

*23E12-AD; 23E10-DBD>UAS-CsChrimson* female flies. ****$P < 0.0001$, n.s. = not significant, $n$ = 24–32 flies per genotype. (**E**) Box plots of locomotor activity counts per minute awake for vehicle-fed flies presented in Fig 1R. Two-way repeated measures ANOVA followed by Sidak's multiple comparisons, n.s. = not significant, $n$ = 25–28 flies per genotype. (**F**) Box plots of loco-motor activity counts per minute awake for retinal-fed flies presented in Fig 1R. Two-way repeated measures ANOVA followed by Sidak's multiple comparisons, n.s. = not significant, $n$ = 24–32 flies per genotype. The raw data underlying parts (A–F) can be found in S1 Data. (TIF)

**S4 Fig. Optogenetic activation of 23E12-AD; 23E10-DBD neurons-male data.** (**A**) Box plots of total sleep change in % ((total sleep on day 3-total sleep on day 2/total sleep on day 2) × 100) for control (Empty-AD; 23E10-DBD) and 23E12-AD; 23E10-DBD male flies expressing CsChrimson upon 627-nm LED stimulation. Two-way ANOVA followed by Sidak's multiple comparisons revealed that total sleep is significantly increased in *23E12-AD; 23E10-DBD>UAS-CsChrimson* male flies stimulated with 627-nm LEDs when compared with vehicle-fed flies. ****$P < 0.0001$, n.s. = not significant, $n$ = 17–32 flies per genotype and condi-tion. (**B**) Box plots of daytime sleep bout duration (in minutes) for vehicle-fed flies presented in (A). Two-way repeated measures ANOVA followed by Sidak's multiple comparisons. *$P < 0.05$, n.s. = not significant, $n$ = 17–32 flies per genotype. (**C**) Box plots of daytime sleep bout duration (in minutes) for retinal-fed flies presented in (A). Two-way repeated measures ANOVA followed by Sidak's multiple comparisons revealed that daytime sleep bout duration is significantly increased in activated *23E12-AD; 23E10-DBD>UAS-CsChrimson* male flies. ****$P < 0.0001$, n.s. = not significant, $n$ = 20–30 flies per genotype. (**D**) Box plots of nighttime sleep bout duration (in minutes) for vehicle-fed flies presented in (A). Two-way repeated mea-sures ANOVA followed by Sidak's multiple comparisons, n.s. = not significant, $n$ = 17–32 flies per genotype. (**E**) Box plots of nighttime sleep bout duration (in minutes) for retinal-fed flies presented in (A). Two-way repeated measures ANOVA followed by Sidak's multiple compari-sons revealed that nighttime sleep bout duration is significantly increased in activated *23E12-AD; 23E10-DBD>UAS-CsChrimson* male flies. **$P < 0.01$, n.s. = not significant, $n$ = 20–30 flies per genotype. (**F**) Box plots of locomotor activity counts per minute awake for vehicle-fed flies presented in (A). Two-way repeated measures ANOVA followed by Sidak's multiple comparisons. *$P < 0.05$, n.s. = not significant, $n$ = 17–32 flies per genotype. (**G**) Box plots of locomotor activity counts per minute awake for retinal-fed flies presented in (A). Two-way repeated measures ANOVA followed by Sidak's multiple comparisons, n.s. = not sig-nificant, $n$ = 20–30 flies per genotype. The raw data underlying parts (A–G) can be found in S1 Data. (TIF)

**S5 Fig. Additional data for TPN1 and VNC-SP neurons.** (**A**) Representative confocal stacks of a female *30A08-LexA>LexAop2-GFP; VT020742-AD; 23E10-DBD>UAS-RFP* fly showing the brain, the VNC, and a magnified view of the cell bodies in the metathoracic ganglion. Yel-low arrows show the TPN1 neurons and their processes in the VNC and in the brain and gray arrows, the VNC-SP neurons and their processes. TPN1 neurons are labeled by 30A08-LexA and VT020742-AD; 23E10-DBD while VNC-SP neurons are only present in VT020742-AD; 23E10-DBD. Green, anti-GFP; magenta, anti-RFP. (**B**) Representative confocal stacks of a female *30A08-LexA>LexAop2-KZip+; VT020742-AD; 23E10-DBD>UAS-GFP* fly showing the brain and the VNC. The KZip+ repressor effectively remove expression in the TPN1 neurons leaving only the VNC-SP neurons. Green, anti-GFP; magenta, anti-nc82 (neuropile marker). (**C**) Box plots of total sleep change in % for *Empty-AD; 23E10-DBD>UAS-CsChrimson*, *VT020742-AD; 23E10-DBD>UAS-CsChrimson*, *Empty-AD; 23E10-DBD>UAS-CsChrimson;*

*30A08-LexA> LexAop2-KZip*+*, and *VT020742-AD; 23E10-DBD>UAS-CsChrimson; 30A08-LexA> LexAop2-KZip*+* vehicle-fed and retinal-fed female flies upon 627-nm LED stimulation. Two-way ANOVA followed by Sidak's multiple comparisons revealed that retinal-fed *VT020742-AD; 23E10-DBD>UAS-CsChrimson* and *VT020742-AD; 23E10-DBD>UAS-CsChrimson; 30A08-LexA> LexAop2-KZip*+* flies increase sleep significantly when stimulated with 627-nm LEDs when compared with vehicle-fed flies. Tukey's multiple comparisons demonstrate that there is no difference in total sleep change between retinal-fed *VT020742-AD; 23E10-DBD>UAS-CsChrimson* and *VT020742-AD; 23E10-DBD>UAS-CsChrimson; 30A08-LexA> LexAop2-KZip*+* flies. **$P < 0.01$, ****$P < 0.0001$, n.s. = not significant, $n = 17$–31 flies per genotype and condition. (**D**) Representative confocal stack images of adult tissues from *VT013602-AD; 23E10-DBD> UAS-mCD8GFP* female flies. GFP is expressed in the VNC, but not the gut, leg, or ovaries. Tissue was dissected, fixed, and stained with DAPI. Green, GFP; blue, DAPI. The raw data underlying part (C) can be found in S1 Data.
(TIF)

**S6 Fig. Optogenetic activation of lines containing only VNC neurons. Additional data. (A)** Box plots of daytime sleep bout duration (in minutes) for retinal-fed female flies presented in Fig 2K. Two-way repeated measures ANOVA followed by Sidak's multiple comparisons indicates that daytime sleep bout duration is increased in retinal-fed *VT020742-AD; 23E10-DBD>UAS-CsChrimson* and *VT013602-AD; 23E10-DBD>UAS-CsChrimson* flies when stimulated with 627-nm LEDs. **$P < 0.01$, ****$P < 0.0001$, n.s. = not significant, $n = 15$–34 flies per genotype. (**B**) Box plots of nighttime sleep bout duration (in minutes) for retinal-fed female flies presented in Fig 2K. Two-way repeated measures ANOVA followed by Sidak's multiple comparisons indicates that nighttime sleep bout duration is increased in retinal-fed *VT020742-AD; 23E10-DBD>UAS-CsChrimson* and *VT013602-AD; 23E10-DBD>UAS-CsChrimson* flies when stimulated with 627-nm LEDs. ***$P < 0.001$, ****$P < 0.0001$, n.s. = not significant, $n = 15$–34 flies per genotype. (**C**) Box plots of total sleep change in % for control (*Empty-AD; 23E10-DBD>UAS-CsChrimson*), *VT020742-AD; 23E10-DBD>UAS-CsChrimson*, *30A08-AD; 23E10-DBD>UAS-CsChrimson* and *VT013602-AD; 23E10-DBD>UAS-CsChrimson* vehicle-fed and retinal-fed male flies upon 627-nm LED stimulation. Two-way ANOVA followed by Sidak's multiple comparisons revealed that retinal-fed *VT020742-AD; 23E10-DBD>UAS-CsChrimson* and *VT013602-AD; 23E10-DBD>UAS-CsChrimson* flies increase sleep significantly when stimulated with 627-nm LEDs when compared with vehicle-fed flies. ****$P < 0.0001$, n.s. = not significant, $n = 18$–40 flies per genotype and condition. (**D**) Box plots of locomotor activity counts per minute awake for retinal-fed flies presented in (C). Two-way repeated measures ANOVA followed by Sidak's multiple comparisons test show that locomotor activity per awake time is increased in *VT020742-AD; 23E10-DBD>UAS-CsChrimson* flies and is not affected in the other genotypes when the flies are stimulated with 627-nm LEDs. **$P < 0.05$, n.s. = not significant, $n = 21$–40 flies per genotype. (**E**) Box plots of daytime sleep bout duration (in minutes) for retinal-fed flies presented in (C). Two-way repeated measures ANOVA followed by Sidak's multiple comparisons indicates that daytime sleep bout duration is increased in retinal-fed *VT020742-AD; 23E10-DBD>UAS-CsChrimson* and *VT013602-AD; 23E10-DBD>UAS-CsChrimson* flies when stimulated with 627-nm LEDs. ****$P < 0.0001$, n.s. = not significant, $n = 21$–40 flies per genotype. (**F**) Box plots of nighttime sleep bout duration (in minutes) for retinal-fed flies presented in (C). Two-way repeated measures ANOVA followed by Sidak's multiple comparisons indicates that nighttime sleep bout duration is increased in retinal-fed *VT020742-AD; 23E10-DBD>UAS-CsChrimson* and *VT013602-AD; 23E10-DBD>UAS-CsChrimson* flies when stimulated with 627-nm LEDs. ****$P < 0.0001$, n.s. = not significant, $n = 21$–40 flies per genotype. The raw data underlying

parts (A–F) can be found in S1 Data.
(TIF)

**S7 Fig. Vehicle-fed sleep data for females in optogenetic activation of lines containing only VNC neurons.** (**A**) Box plots of locomotor activity counts per minute awake for vehicle-fed flies presented in Fig 2K. Two-way repeated measures ANOVA followed by Sidak's multiple comparisons test found that locomotor activity per awake time is reduced in vehicle-fed *VT020742-AD; 23E10-DBD>UAS-CsChrimson* and *VT013602-AD; 23E10-DBD>UAS-Cs-Chrimson* flies that are stimulated with 627-nm LEDs. ${}^{**}P < 0.01$, ${}^{***}P < 0.001$, n.s. = not significant, $n$ = 13–32 flies per genotype. (**B**) Box plots of daytime sleep bout duration for vehicle-fed flies presented in Fig 2K. Two-way repeated measures ANOVA followed by Sidak's multiple comparisons test found no difference in daytime sleep bout duration when the flies are stimulated with 627-nm LEDs, n.s. = not significant, $n$ = 13–32 flies per genotype. (**C**) Box plots of nighttime sleep bout duration for vehicle-fed flies presented in Fig 2K. Two-way repeated measures ANOVA followed by Sidak's multiple comparisons test show that nighttime sleep bout duration is not different when vehicle-fed flies are stimulated with 627-nm LEDs, n.s. = not significant, $n$ = 13–32 flies per genotype. The raw data underlying parts (A–C) can be found in S1 Data.
(TIF)

**S8 Fig. Vehicle-fed sleep data for males in optogenetic activation of lines containing only VNC neurons.** (**A**) Box plots of locomotor activity counts per minute awake for vehicle-fed flies presented in S6C Fig. Two-way repeated measures ANOVA followed by Sidak's multiple comparisons test found that locomotor activity per awake time is reduced in vehicle-fed *VT020742-AD; 23E10-DBD>UAS-CsChrimson* and *VT013602-AD; 23E10-DBD>UAS-Cs-Chrimson* flies that are stimulated with 627-nm LEDs. ${}^{**}P < 0.01$, n.s. = not significant, $n$ = 18–32 flies per genotype. (**B**) Box plots of daytime sleep bout duration for vehicle-fed flies presented in S6C Fig. Two-way repeated measures ANOVA followed by Sidak's multiple comparisons test found no difference in daytime sleep bout duration when the flies are stimulated with 627-nm LEDs, n.s. = not significant, $n$ = 18–32 flies per genotype. (**C**) Box plots of nighttime sleep bout duration for vehicle-fed flies presented in S6C Fig. Two-way repeated measures ANOVA followed by Sidak's multiple comparisons test show that nighttime sleep bout duration is not increased when vehicle-fed flies are stimulated with 627-nm LEDs, n.s. = not significant, $n$ = 18–32 flies per genotype. The raw data underlying parts (A–C) can be found in S1 Data.
(TIF)

**S9 Fig. Optogenetic activation multibeam data.** (**A**) Box plots of daytime sleep bout duration for multibeam analysis of vehicle-fed flies presented in Fig 2M. Two-way repeated measures ANOVA followed by Sidak's multiple comparisons test found no difference in daytime sleep bout duration when the flies are stimulated with 627-nm LEDs, n.s. = not significant, $n$ = 19–24 flies per genotype. (**B**) Box plots of daytime sleep bout duration for multibeam analysis of retinal-fed flies presented in Fig 2M. Two-way repeated measures ANOVA followed by Sidak's multiple comparisons test show that daytime sleep bout duration is significantly increased in *VT013602-AD; 23E10-DBD>UAS-CsChrimson* flies stimulated with 627-nm LEDs. ${}^{***}P < 0.001$, n.s. = not significant, $n$ = 23–24 flies per genotype. (**C**) Box plots of nighttime sleep bout duration for multibeam analysis of vehicle-fed flies presented in Fig 2M. Two-way repeated measures ANOVA followed by Sidak's multiple comparisons test. ${}^{*}P < 0.05$, n.s. = not significant, $n$ = 19–24 flies per genotype. (**D**) Box plots of nighttime sleep bout duration for multibeam analysis of retinal-fed flies presented in Fig 2M. Two-way repeated measures

ANOVA followed by Sidak's multiple comparisons test, n.s. = not significant, $n = 23–24$ flies per genotype. The raw data underlying parts (A–D) can be found in S1 Data.
(TIF)

**S10 Fig. Optogenetic activation of VNC-SP neurons increases sleep, not feeding and grooming behaviors.** (**A**) Diagram of a multibeam tube with 17 individual infrared beams in red and fly for scale. (**B**) Graph of average fly beam position over 24 h. Each number on the y-axis represents a beam with the food positioned at 0. Beam position over 24 h was averaged at baseline with LED OFF (white bars) and activation day with LED ON (gray bars) for control *Empty-AD; 23E10-DBD>UAS-CsChrimson* and *VNC-SP>UAS-CsChrimson* flies fed vehicle or retinal. Three-way ANOVA followed by Tukey's multiple comparisons test found no difference between the control and experimental fly position, n.s. = not significant, $n = 19–24$ flies per genotype and condition. (**C**) Video analysis of control and *VNC-SP>UAS-CsChrimson* female flies fed vehicle or retinal. Recording was performed at ZT1-2 for 1 h on consecutive days at baseline (LED OFF) and activation (LED ON). Behaviors were manually scored, and amount of time spent on each behavior over the hour is shown. Three-way ANOVA followed by Tukey's multiple comparisons test found that *VNC-SP>UAS-CsChrimson* flies fed retinal with LED ON sleep significantly more than controls, ****$P < 0.0001$, but spend less time walking (****$P < 0.0001$), grooming (**$P < 0.01$), or feeding (*$P < 0.05$). Rest (periods of inactivity shorter than 5 min) are not different, n.s. = not significant, $n = 5–7$ flies for each genotype and condition. The raw data underlying parts (B) and (C) can be found in S1 Data.
(TIF)

**S11 Fig. VNC-SP neurons are present in many sleep-promoting Split-GAL4 lines.** (**A**) Representative confocal stacks of a female *VT013602-AD, VT020742-AD; 23E10-DBD>UAS-GFP* fly showing the brain and the VNC. Gray arrows show the processes of VNC-SP neurons in the brain. Green, anti-GFP; magenta, anti-nc82 (neuropile marker). (**B**) Representative confocal stacks of a female *VT013602-AD; VT020742-DBD>UAS-GFP* fly showing the brain and the VNC. Gray arrows show the processes of VNC-SP neurons in the brain. Green, anti-GFP; magenta, anti-nc82 (neuropile marker). (**C**) Sleep profile in minutes of sleep per hour for *Empty-AD; Empty-DBD>UAS-TrpA1* female flies maintained at 22°C (blue line) and transferred to 31°C (red line). (**D**) Sleep profile in minutes of sleep per hour for *VT013602-AD; VT020742-DBD>UAS-TrpA1* female flies maintained at 22°C (blue line) and transferred to 31°C (red line). (**E**) Box plots of total sleep change in % for female control (Empty-AD; Empty-DBD) and VT013602-AD; VT020742-DBD flies expressing UAS-TrpA1. A two-tailed unpaired *t* test revealed that activating VT013602-AD; VT020742-DBD neurons significantly increases sleep compared with controls. ****$P < 0.0001$, $n = 13–14$ flies per genotype. (**F**) Representative confocal stacks of a female *23E12-AD; VT020742-DBD>UAS-GFP* fly showing the brain and the VNC. Gray arrows show the processes of VNC-SP neurons in the brain. Green, anti-GFP; magenta, anti-nc82 (neuropile marker). (**G**) Representative confocal stacks of a female *23E12-AD; VT013602-DBD>UAS-GFP* fly showing the brain and the VNC. Gray arrows show the processes of VNC-SP neurons in the brain. Green, anti-GFP; magenta, anti-nc82 (neuropile marker). (**H**) Box plots of total sleep change in % for female control (Empty-AD; Empty-DBD), 23E12-AD; VT020742-DBD, and 23E12-AD; VT013602-DBD flies expressing UAS-TrpA1. A one-way ANOVA followed by Tukey's multiple comparisons demonstrate that activating neurons contained in the 23E12-AD; VT020742-DBD and 23E12-AD; VT013602-DBD line significantly increases sleep compared with controls. ***$P < 0.001$, ****$P < 0.0001$, $n = 12–14$ flies per genotype. The raw data underlying parts (E) and (H) can be found in S1 Data.
(TIF)

**S12 Fig. Silencing VNC-SP neurons-additional data.** (**A**) Box plots of daytime sleep bout duration (in minutes) for flies presented in Fig 3A. Two-tailed Mann–Whitney U tests revealed that daytime sleep bout duration is significantly reduced in *VNC-SP>Kir2.1* female flies compared to controls. $^{***}P < 0.001$, $n$ = 58–60 flies per genotype. (**B**) Box plots of nighttime sleep bout duration (in minutes) for flies presented in Fig 3A. Two-tailed Mann–Whitney U tests revealed that nighttime sleep bout duration is significantly reduced in *VNC-SP>Kir2.1* female compared to controls. $^{***}P < 0.001$, $n$ = 58–60 flies per genotype. (**C**) Box plots of daytime sleep bout duration (in minutes) analyzed with the multibeam system for flies presented in Fig 3D. Two-tailed Mann–Whitney U tests revealed that daytime sleep bout duration is not different between *VNC-SP>Kir2.1* female flies and controls, n.s. = not significant, $n$ = 28–31 flies per genotype. (**D**) Box plots of nighttime sleep bout duration (in minutes) analyzed with the multibeam system for flies presented in Fig 3D. Two-tailed Mann–Whitney U tests revealed that nighttime sleep bout duration is significantly reduced in *VNC-SP>Kir2.1* female compared to controls. $^{**}P < 0.01$, $n$ = 28–31 flies per genotype. (**E**) Sleep profile in minutes of sleep per hour for control (*Empty-AD; 23E10-DBD>UAS-Kir2.1*, blue line) and *VNC-SP>Kir2.1* (*VT013602-AD; 23E10-DBD>UAS-Kir2.1*, red line) male flies. (**F**) Box plots of total sleep time (in minutes) for flies presented in (E). A two-tailed Mann–Whitney U test revealed that total sleep is significantly reduced in *VNC-SP>Kir2.1* male flies compared to controls. $^{****}P < 0.0001$, $n$ = 48 flies per genotype. (**G**) Box plots of daytime sleep bout duration (in minutes) for flies presented in (E). Two-tailed Mann–Whitney U tests revealed that daytime sleep bout duration is significantly reduced in *VNC-SP>Kir2.1* and male flies compared to controls. $^{*}P < 0.05$, $n$ = 48 flies per genotype. (**H**) Box plots of nighttime sleep bout duration (in minutes) for flies presented in (E). Two-tailed Mann–Whitney U tests revealed that nighttime sleep bout duration is significantly reduced in *VNC-SP>Kir2.1* male flies compared to controls. $^{***}P < 0.001$, $n$ = 48 flies per genotype. (**I**) Box plots of locomotor activity counts per minute awake for flies presented in (E). Two-tailed Mann–Whitney U tests revealed that activity per minute awake is unchanged in *VNC-SP>Kir2.1* male flies compared to controls, n.s. = not significant, $n$ = 48 flies per genotype. (**J**) Sleep profile in minutes of sleep per hour for control (*Empty-AD; 23E10-DBD>UAS-Shi^{ts1}*) male flies at 22°C (blue line) and 32°C (red line). (**K**) Sleep profile in minutes of sleep per hour for *VNC-SP>UAS-Shi^{ts1}* male flies at 22°C (blue line) and 32°C (red line). (**L**) Box plots of nighttime sleep change in % for male flies presented in (J) and (K). Two-tailed unpaired $t$ tests revealed that *VNC-SP>Shi^{ts1}* flies lose significantly more sleep when transferred to 32°C compared with controls. $^{*}P < 0.05$, $n$ = 31 flies per genotype. The raw data underlying parts (A–D), (F–I), and (L) can be found in S1 Data. (TIF)

**S13 Fig. Additional data demonstrating that VNC-SP neurons are cholinergic.** (**A**) Representative confocal stacks of a female *ChAT-LexA>LexAop2-GFP; VT013602-AD; 23E10-DBD (VNC-SP)>UAS-RFP* focusing on the cell bodies in the metathoracic ganglion of the VNC. Gray arrows show the VNC-SP neurons that are co-labeled by the ChAT-LexA driver. Green, anti-GFP; magenta, anti-RFP. (**B**) Representative confocal stacks of a female *ChAT-LexA>LexAop2-GFP; 23E10-GAL4>UAS-RFP*. Top panels, focusing on the cell bodies in the metathoracic ganglion of the VNC. Gray arrows show the VNC-SP neurons that are co-labeled by the ChAT-LexA driver. Bottom panels, focusing on dFB neurons. Red arrows show 23E10-GAL4 dFB neurons that are also labeled by ChAT-LexA. Green, anti-GFP; magenta, anti-RFP. (**C**) Quantification of ChAT knockdown efficiency by RNAi. qPCR was performed on the body and eyes of control flies (*GMR-GAL4>+*) and flies expressing ChAT RNAi (line 60028 and line 25856) in the eyes driven by GMR-GAL4 (*GMR-GAL4>ChAT RNAi*).

Expression levels were normalized to control levels. Two-way ANOVA followed by Sidak's multiple comparisons revealed that in *GMR-GAL4>ChAT RNAi 60028* flies, ChAT levels are significantly reduced in the eyes but not in the body compared with controls. $^{**}P < 0.01$, n.s. = not significant, $n = 3–4$ replicates per genotype. (**D**) ChAT immunostaining in *VNC-SP>UAS-GFP* (top panels) and *VNC-SP>UAS-GFP; UAS-ChAT RNAi* flies (bottom panels). Gray arrows point to neurons positive for GFP and ChAT. Green, anti-GFP; magenta, anti-ChAT. (**E**) Quantification of data presented in (D). A one-tailed unpaired *t* test revealed that expressing ChAT RNAi in VNC-SP neurons significantly reduce ChAT levels as measured with ChAT antibody staining. $^{*}P < 0.05$, $n = 14–17$ VNC analyzed per genotype. (**F**) Box plots of total sleep time (in minutes) for control (*Empty-AD; 23E10-DBD>ChAT 60028*) and *VNC-SP>ChAT 60028* male flies. A two-tailed Mann–Whitney U test revealed that *VNC-SP>ChAT^RNAi* male flies sleep significantly less than controls. $^{****}P < 0.0001$, $n = 24–30$ flies per genotype. (**G**) Box plots of daytime sleep bout duration (in minutes) for control (*Empty-AD; 23E10-DBD>ChAT 60028*) and *VNC-SP>ChAT 60028* male flies. A two-tailed Mann–Whitney U test revealed that *VNC-SP> ChAT^RNAi* male flies daytime sleep bout duration is significantly reduced compared with controls. $^{***}P < 0.001$, $n = 24–30$ flies per genotype. (**H**) Box plots of nighttime sleep bout duration (in minutes) for control (*Empty-AD; 23E10-DBD>ChAT 60028*) and *VNC-SP>ChAT 60028* male flies. A two-tailed Mann–Whitney U test revealed that *VNC-SP> ChAT^RNAi* male flies nighttime sleep bout duration is significantly reduced compared with controls. $^{***}P < 0.001$, $n = 24–30$ flies per genotype. (**I**) Box plots of locomotor activity counts per minute awake for control (*Empty-AD; 23E10-DBD>ChAT 60028*) and *VNC-SP>ChAT 60028* male flies. A two-tailed unpaired *t* test revealed that there is no difference in waking activity between *VNC-SP> ChAT^RNAi* male flies and controls, n.s. = not significant, $n = 24–30$ flies per genotype. (**J**) Box plots of total sleep time (in minutes) for control (*Empty-AD; 23E10-DBD>ChAT 25856*) and *VNC-SP>ChAT 25856* female flies. A two-tailed unpaired *t* test revealed that *VNC-SP> ChAT 25856* female flies sleep significantly less than controls. $^{*}P < 0.05$, $n = 27–28$ flies per genotype. (**K**) Box plots of daytime sleep bout duration (in minutes) for control (*Empty-AD; 23E10-DBD>ChAT 25856*) and *VNC-SP>ChAT 25856* female flies. A two-tailed Mann–Whitney U test revealed no difference in daytime sleep bout duration, n.s. = not significant, $n = 27–28$ flies per genotype. (**L**) Box plots of nighttime sleep bout duration (in minutes) for control (*Empty-AD; 23E10-DBD>ChAT 25856*) and *VNC-SP>ChAT 25856* female flies. A two-tailed Mann–Whitney U test revealed no difference in nighttime sleep bout duration, n.s. = not significant, $n = 27–28$ flies per genotype. (**M**) Box plots of locomotor activity counts per minute awake for control (*Empty-AD; 23E10-DBD>ChAT 25856*) and *VNC-SP>ChAT 25856* female flies. A two-tailed Mann–Whitney U test revealed that *VNC-SP>ChAT 25856* flies are more active than controls when awake. $^{***}P < 0.001$, $n = 27–28$ flies per genotype. (**N**) Box plots of total sleep time (in minutes) for control (*Empty-AD; 23E10-DBD>ChAT 25856*) and *VNC-SP>ChAT 25856* male flies. A two-tailed unpaired *t* test revealed no difference in total sleep time, n.s. = not significant, $n = 30–47$ flies per genotype. (**O**) Box plots of daytime sleep bout duration (in minutes) for control (*Empty-AD; 23E10-DBD>ChAT 25856*) and *VNC-SP>ChAT 25856* male flies. A two-tailed Mann–Whitney U test revealed no difference in daytime sleep bout duration, n.s. = not significant, $n = 30–47$ flies per genotype. (**P**) Box plots of nighttime sleep bout duration (in minutes) for control (*Empty-AD; 23E10-DBD>ChAT 25856*) and *VNC-SP>ChAT 25856* male flies. A two-tailed Mann–Whitney U test revealed no difference in nighttime sleep bout duration, n.s. = not significant, $n = 30–47$ flies per genotype. (**Q**) Box plots of locomotor activity counts per minute awake for control (*Empty-AD; 23E10-DBD>ChAT 25856*) and *VNC-SP>ChAT 25856* male flies. A two-tailed unpaired *t* test revealed no difference in waking activity, n.s. = not significant, $n = 30–47$

flies per genotype. (**R**) Box plots of total sleep change in % for *VT013602-AD; 23E10-DBD>UAS-CsChrimson; 2x LexAop2KZip⁺* (control, no LexA driver) and for *VT013602-AD; 23E10-DBD>UAS-CsChrimson; ChAT-LexA>2x LexAop2KZip⁺* vehicle-fed and retinal-fed female flies upon 627-nm LED stimulation. Two-way ANOVA followed by Sidak's multiple comparisons revealed that retinal-fed *VT013602-AD; 23E10-DBD>UAS-Cs-Chrimson; ChAT-LexA>2x LexAop2KZip⁺* flies do not increase sleep compared with vehicle-fed flies when stimulated with 627-nm LEDs. Control flies increase sleep significantly. ****$P < 0.0001$, n.s. = not significant, $n = 16–27$ flies per genotype and condition. (**S**) Representative confocal stacks of *VT013602-AD; 23E10-DBD>UAS-GFP* (left) and *VT013602-AD; 23E10-DBD>UAS-GFP; ChAT-LexA>2x LexAop2KZip⁺* (right) female flies. Expression of GFP in VNC-SP neurons and in their bowtie brain processes is completely abolished by the expression of the KZip⁺ repressor. Gray arrows show VNC-SP neurons. Green, anti-GFP; magenta, anti-RFP. The raw data underlying parts (C) and (E–R) can be found in S1 Data. (TIF)

**S14 Fig. Removing VNC-SP neurons from the expression pattern of the 23E12-AD; 23E10-DBD and VT020742-AD; 23E10-DBD Split-GAL4 lines blocks sleep promotion.** (**A**) Representative confocal stacks of a female *23E12-AD; 23E10-DBD>UAS-GFP* fly showing the brain and the VNC. The gray rectangle shows the processes of VNC-SP neurons in the brain. Gray arrows show the cell bodies of VNC-SP neurons. Yellow arrows show the TPN1 neurons. Green, anti-GFP; magenta, anti-ChAT. (**B**) Representative confocal stacks of a female *23E12-AD; 23E10-DBD>UAS-GFP; ChAT-LexA> LexAop2KZip⁺* fly showing the brain and the VNC. The gray rectangle highlights the absence of processes of VNC-SP neurons in the brain. Yellow arrows show the TPN1 neurons. Green, anti-GFP. (**C**) Representative confocal stacks of a female *VT020742-AD; 23E10-DBD>UAS-GFP* fly showing the brain and the VNC. The gray arrows show the processes of VNC-SP neurons in the brain. Green, anti-GFP. (**D**) Representative confocal stacks of a female *VT020742-AD; 23E10-DBD>UAS-GFP; ChAT-LexA> LexAop2KZip⁺* fly showing the brain and the VNC. Green, anti-GFP. (**E**) Box plots of total sleep change in % for female control (Empty-AD; Empty-DBD), VT013602-AD; 23E10-DBD, VT020742-AD; 23E10-DBD, and 23E12-AD; 23E10-DBD flies expressing UAS-TrpA1 or UAS-TrpA1; ChAT-LexA> LexAop2KZip⁺. A one-way ANOVA followed by Tukey's multiple comparisons demonstrate that activating neurons contained in the 3 Split-GAL4 lines significantly increases sleep compared with controls and that removing expression in the VNC-SP neurons by expressing the KZip⁺ repressor in ChAT expressing neurons blocks this sleep increase. ****$P < 0.0001$, n.s. = not significant, $n = 14–42$ flies per genotype. The raw data underlying part (E) can be found in S1 Data. (TIF)

**S15 Fig. VNC-SP neurons are well positioned to integrate sensory inputs in the VNC and send this information to the brain.** Representative confocal stacks of a female *VT013602-AD; 23E10-DBD> UAS-5xDenMark-V5; UAS-syt.eGFP*. Top panels, focusing on the brain "bowtie" processes. Middle panels, the VNC and bottom panels, a magnified view of the cell bodies area. Green, anti-GFP; magenta, anti-V5. (TIF)

**S1 Table. Quantification of the number of cells labeled by the 23E10-GAL4 driver.** Average number ± SEM and range of dFB neurons, all brain neurons and VNC neurons labeled in *23E10-GAL4>UAS-GFP* female flies. The raw data underlying this table can be found in S1 Data. (TIF)

**S2 Table. List of fly stocks used in this study.**
(DOCX)

**S1 Movie. Brain of a 23E10-GAL4>UAS-GFP female.**
(MP4)

**S2 Movie. VNC of a 23E10-GAL4>UAS-GFP female.**
(MP4)

**S3 Movie. Brain of a 23E12-AD; 23E10-DBD>UAS-GFP female.**
(MP4)

**S4 Movie. VNC of a 23E12-AD; 23E10-DBD>UAS-GFP female.**
(MP4)

**S5 Movie. Brain of a VT013602AD-23E10-DBD (VNC-SP)>UAS-GFP female.**
(MP4)

**S6 Movie. VNC of a VT013602AD-23E10-DBD (VNC-SP)>UAS-GFP female.**
(MP4)

**S1 Data. Raw data underlying all Figs 1–5 and S2–S14 and S1 Table.**
(XLSX)

## Acknowledgments

We thank Aaron DiAntonio, Gerry Rubin, Paul Shaw, and Jeff Price for sharing reagents. We also thank Krishna Melnattur and Samuel Bouyain for comments on this manuscript.

## Author Contributions

**Conceptualization:** Stephane Dissel.

**Funding acquisition:** Stephane Dissel.

**Investigation:** Joseph D. Jones, Brandon L. Holder, Kiran R. Eiken, Alex Vogt, Adriana I. Velarde, Alexandra J. Elder, Jennifer A. McEllin, Stephane Dissel.

**Methodology:** Joseph D. Jones, Jennifer A. McEllin, Stephane Dissel.

**Project administration:** Stephane Dissel.

**Supervision:** Stephane Dissel.

**Visualization:** Joseph D. Jones, Jennifer A. McEllin, Stephane Dissel.

**Writing – original draft:** Joseph D. Jones, Brandon L. Holder, Jennifer A. McEllin, Stephane Dissel.

**Writing – review & editing:** Joseph D. Jones, Brandon L. Holder, Jennifer A. McEllin, Stephane Dissel.

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
