## [Editor Report · Decision Letter 0]

13 May 2022

Dear Dr Dissel, 

Thank you for submitting your manuscript entitled "Regulation of sleep by cholinergic neurons located outside the central brain in Drosophila." for consideration as a Research Article by PLOS Biology.

Your manuscript has now been evaluated by the PLOS Biology editorial staff and I am writing to let you know that we would like to send your submission out for external peer review.

Once your full submission is complete, your paper will undergo a series of checks in preparation for peer review. Once your manuscript has passed the checks it will be sent out for review. To provide the metadata for your submission, please Login to Editorial Manager (https://www.editorialmanager.com/pbiology) within two working days, i.e. by May 15 2022 11:59PM.

If your manuscript has been previously reviewed at another journal, PLOS Biology is willing to work with those reviews in order to avoid re-starting the process. Submission of the previous reviews is entirely optional and our ability to use them effectively will depend on the willingness of the previous journal to confirm the content of the reports and share the reviewer identities. Please note that we reserve the right to invite additional reviewers if we consider that additional/independent reviewers are needed, although we aim to avoid this as far as possible. In our experience, working with previous reviews does save time. 

If you would like to send previous reviewer reports to us, please email me at kdickson@plos.org to let me know, including the name of the previous journal and the manuscript ID the study was given, as well as attaching a point-by-point response to reviewers that details how you have or plan to address the reviewers' concerns. 

Kind regards,

Kris

Kris Dickson, Ph.D (she/her)

Neurosciences Senior Editor/Section Manager

PLOS Biology

kdickson@plos.org

---

## [Decision Letter · Decision Letter 1]

28 Jun 2022

Dear Dr Dissel,

Thank you for your patience while your manuscript "Regulation of sleep by cholinergic neurons located outside the central brain in Drosophila." was peer-reviewed at PLOS Biology. Your manuscript has been evaluated by the PLOS Biology editors, an Academic Editor with relevant expertise, and by several independent reviewers.

You can find the reviewers' detailed feedback at the end of this email. In discussion with our Academic Editor on these reviews, they expressed some level of overlapping concerns to those raised by Reviewer 3 regarding your behavioural characterization, particularly in light of the published data describing an opposite effect noted by Reviewer 3. However, we also appreciate the potential difficulties in repeating the key experiments with a video tracking system if one is not already set up in your laboratory. We therefore ask that you attempt to use some alternative methods either currently available to you or via collaboration with another laboratory to allow activity monitoring with higher resolution. Our aim is to ensure you are able to gain a finer-grained description of the times spent on specific aspects of a behavior to help confirm the key behavioral findings in your work, and to separate out behaviors that your current system might inappropriately qualify as sleep (e.g. grooming, feeding, etc). In addition, our Academic Editor particularly noted the need to include the splitGal4 screen data.

If you are willing to take on the extra work involved to address these points and the other reviewers' concerns, and we are willing to relax our standard revision time to allow you 6 months to revise your study. Please email us (plosbiology@plos.org) if you have any questions or concerns, or envision needing a (short) extension. 

Please note that we would be looking to see that your revisions convince us, the Academic Editor and the reviewers. We also appreciate that the scale of the requested additional work is significant. If you have any questions about these revisions, please feel free to reach out with a revision plan that we would be happy to discuss with our Academic Editor.

**IMPORTANT - SUBMITTING YOUR REVISION**

*Resubmission Checklist*

*Published Peer Review*

*PLOS Data Policy*

*Blot and Gel Data Policy*

Sincerely,

Kris

Kris Dickson, Ph.D. (she/her)

Neurosciences Senior Editor/Section Manager

PLOS Biology

kdickson@plos.org

REVIEWS:

Reviewer's Responses to Questions

PLOS authors have the option to publish the peer review history of their article (what does this mean?). If published, this will include your full peer review and any attached files.

Reviewer #1: No

Reviewer #2: No

Reviewer #3: Yes: Giorgio Gilestro

Reviewer #1: This study entails further mapping of the sleep system in Drosophila. The authors build upon a previous sleep system transgenic tool, 23E10-GAL4, which is expressed in fan-body neurons in the brain. By developing intersectional transgenic tools, Split-GAL4, they find that a small subset of neurons located outside the brain, in the ventral nerve cord, are also involved in sleep control. They go on to map their neurotransmitter expression and their basic neuroanatomy, and provide some evidence that the fan-body and the ventral nerve cord neurons control different aspects of sleep. Their analysis appears sound and the text is largely free from overinterpretations. Hence, there is no question that this study will be of great value for Drosophila sleep research. However, against the backdrop of the immense body of work on mammalian sleep, and given that sleep in mammals is also known to engage both brain and spinal cord circuits, it is unclear if the finding of ventral nerve cord sleep neurons in the fly is surprising and novel enough to appeal to the broad readership of PLoS Biology. A deeper understanding of the specific role of the ventral nerve cord sleep neurons in sleep control would have greatly strengthened the paper. In addition, there are some minor things that could improve this piece.

Introduction: The description of the fly sleep system is confusing, for instance rows 61-62 "Importantly, neurons that project to the dFB have emerged as a key sleep-regulating area." I would suggest adding a cartoon to figure 1 outlining what is known about the sleep system in flies. 

Abstract: The abstract is perhaps a bit too "23E10-GAL4-centered". The main finding of this study is that they identify neurons in the nerve cord that are involved in sleep, not that they are a subset of 23E10-GAL4-expressing neurons, which one would gather is only meaningful to a handful of labs around the world. The same applies to the Introduction, rows 81-84.

Introduction and/or Results, first section: It would be good if they define how they measure fly sleep. It's outlined in the Methods, but should be described here as well.

Results, rows 165-166: They should provide some info as to how they arrived at the intersectional Split-GAL4 tool used herein.

Fig 5: Outline cells in B and C as well.

Reviewer #2: All in all this is very comprehensive and incisive study leading to the identification of a precise pair of neurons in the Drosophila CNS, but outside the brain, which has a clear role in sleep promotion but not in sleep rebound. The experimental work is rigorous and the logic generally sound. I think the advance is clear and two-fold: first it provides a stark example of the limitations of "looking where the light is". Meaning if you only look in the brain, you will focus on brain neurons only. The second advance is to identify a non-brain neuronal cell class that potently affects sleep state (not waking state) in both loss of function and gain of function conditions. As suggested by the authors, it implies significant flow of information from the periphery to the CNS to influence the movement into sleep. This work should have a substantial influence on the sleep field (but also other fields) in the fly model system. I have only minor comments that may help improve this excellent manuscript.

Line 79

The authors write:

"Strikingly…"

Suggest removing such descriptors and leaving impressions about the information for the reader to form.

Line 81

The authors write:

"Thus, our work reveals that an extensively used "dFB-specific" GAL4 driver contains non-dFB sleep-promoting neurons, raising important questions about the role of the dFB in sleep regulation and highlighting the importance of using specific tools in behavioral manipulations."

These are important considerations and I agree the Gal4 lines contains more elements than the dFB that appear to promote sleep. However, I'm not sure I understand why this results calls into question the role of the dFB in sleep regulation? Further, beyond using three different drivers that shared dFB expression, the original report (Donlea et al) made the same conclusion:

"In contrast, the expression pattern of these three drivers did not show prominent overlap in neurons in other brain regions (fig. S2) (12). Thus, although we cannot formally exclude a role for brain regions outside the dorsal FB, it is likely that the dorsal FB plays a role in regulating quiescence." 

The present work therefore adds important new information that was not previously excluded; it does not disprove conclusions previously expressed. I suggest the language be modified to reflect that more accurately.

line 150

The authors write:

"To clarify the expression pattern of 23E10-GAL4, we expressed GFP under its control and identified more than 50 GFP-positive neurons in the brain, only half of which are dFB neurons (Fig 1G and 1H, and Table S1). In addition, 23E10-GAL4 expresses in about 18 neurons in the VNC …"

Was expression outside the CNS examined?

line 165 

The authors write: 

"The description and results of this Split-GAL4 screen are beyond the scope of this manuscript and will be described elsewhere."

I recommend providing a description of the screen with this report. Publishing has moved beyond the days of including "unpublished results".

line 194

The authors write

"Two of these neurons show a specific pattern of expression, with brain processes located in the superior medial protocerebrum …:

Are the bowtie processes in the protocerebrum visible in 23E10-Gal4>GFP brains?

line 325

The authors write:

"Optogenetic activation of 30A08-AD; 23E10-DBD neurons has no effect on sleep (Fig 3K-3N 326 and Fig S4A-S4D), indicating that the sleep-promoting capacity of VT020742-AD; 23E10-DBD 327 most likely lies within the two "bowtie" neurons."

Strictly speaking, it says TPN1 neurons are not sufficient. It says nothing about the sufficiency of the bowtie neurons. The experiments described later on line 333 do provide that demonstration, so I recommend modifying the text in lines 325-327.

line 467

The authors write:

"…RNAi lines for ChAT. Out of these two lines, only one (60028) led to a significant reduction in 468 ChAT levels, as judged by qPCR and immunocytochemistry, and was used in our behavioral 469 studies (Fig S7C-S7E)."

It would be useful to assess behavior with the 2nd RNAi to ask if there is a correlation. I do not like to ask for additional experiments generally, but this one appears of limited scope. I could strengthen the conclusion linking VNC-SP action via cholinergic transmission.

Reviewer #3: This is an excellent piece of work and well presented. The field has reached dangerous level of sloppiness and papers like this one are a stark reminder that we all ought to be better. The genetics is fantastic and the data presentation is careful. 

There are only two comments that I need to make. 

Major comment (decision left to the editor).

The authors show how important it is to be careful and considerate in drawing conclusions based on approximate genetics. They do an excellent job in demonstrating that 23E10 neurons include a small but important population of activity regulating neurons in the VNC. They, however, take a suboptimal approach themselves when it comes to behavioural characterization, relying exclusively on the DAM system. The DAM system is obsolete. It should no longer be used for ANY sleep paper, let alone for a beautiful paper like this one whose purpose is to correct other people's work. I would like to see the key experiments reproduced using a more appropriate system. My recommendation would be to employ video tracking or, as a second best , a more modern multi-beam system.

My lab recently published an experiment very similar to the one shown in Figure 1BCD of this paper using ethoscope-based videotracking (see French et al Nature 2021, Supplementary Figure 5 - the only difference is the activation temperature 29C vs 31C). In that experiment, we showed that 23E10 activation using trpA1 does not lead to an increase in sleep but a decrease in sleep caused by what is likely to be an increase in feeding. Keep in mind that any manipulation that increases feeding activity will be misunderstood as an increase in sleep by the DAM system because it will result in less beam crossing activity. I commend the fact that you at least use arousal threshold as a way to improve your findings (Fig 3O) but be aware that arousal threshold will be changed by feeding activity too (a fly won't necessarily react to mechanical stimuli while it's eating).

Minor comments (decision left to the authors).

1. Personally, I do not place any value in any analysis of sleep bouts (duration, numbers) when data are collected with DAMs because they simply cannot offer enough resolution for this kind of secondary analysis. This paper is full of those so if things must stay as they are I would recommend moving them to supplementary. That would also allow to join Figure 1 and 2 into one figure. I think, in general, the paper can benefit from being shortened. It has a simple message that gets a bit lost in a long narrative.

2.The combination of split GAL4 line is difficult to follow and one will easily get lost. You can solve this in multiple ways: 1) having cartoons showing "which line expresses where" in Figure 3; 2) giving a human-readable nickname to the split GAL4 combinations, in the figure and the text. I recommend doing both.

3. If you invert the position of panels A and BC in figure 6 then you can remove the question mark from your figure and give more weight to your findings.

Giorgio Gilestro, Imperial College London

---

## [Decision Letter · Decision Letter 2]

10 Jan 2023

Dear Dr Dissel,

Thank you for your patience while we considered your revised manuscript "Regulation of sleep by cholinergic neurons located outside the central brain in Drosophila." for publication as a Research Article at PLOS Biology. This revised version of your manuscript has been evaluated by the PLOS Biology editors, the Academic Editor and the original reviewers.

Based on the reviews and our Academic Editor's assessment of your revision, we are likely to accept this manuscript for publication. Before moving forward however, we ask that you address the remaining minor concerns raised by Reviewer 2 and that you include a model/cartoon figure depicting the sleep system in flies as was originally requested by Reviewer 1. We find that such summary figures can be quite helpful for our broad readership, as was also noted by Reviewer 1. 

Please also make sure to address all of the data and other policy-related requests listed below my signature. Note that failure to fully address these points will lead to delays in processing your submission further.

We expect to receive your revised manuscript within two weeks. 

*Published Peer Review History*

*Press*

Sincerely,

Kris

Kris Dickson, Ph.D., (she/her)

Neurosciences Senior Editor/Section Manager,

kdickson@plos.org,

PLOS Biology

FUNDING:

Please include any relevant grant number(s) for the UMKC start-up fund as applicable. 

DATA POLICY:

You may be aware of the PLOS Data Policy, which requires that all data be made available without restriction: http://journals.plos.org/plosbiology/s/data-availability. For more information, please also see this editorial: http://dx.doi.org/10.1371/journal.pbio.1001797. Note that we do not require all raw data. Rather, we ask that all individual quantitative observations that underlie the data summarized in the figures and results of your paper be made available.

Thank you for supplying this metadata in the accompanying "Metadata Jones Plos biology resubmission" file. 

***When revising your study, please ensure that figure legends in your manuscript include statements directing readers to this file so that it is clear where the underlying data can be found, and ensure your supplemental data file/s has a legend.

***Please ensure that your Data Statement in the submission system accurately describes where your data can be found.

DATA NOT SHOWN?

- Please note that per journal policy, we do not allow the mention of "data not shown", "personal communication", "manuscript in preparation" or other references to data that is not publicly available or contained within this manuscript. Please check your submission carefully for any such statements and either remove mention of any such data or provide figures presenting the results and the data underlying the figure(s).

Reviewer remarks:

Do you want your identity to be public for this peer review?

Reviewer #1: No

Reviewer #2: No

Reviewer #3: No

Reviewer #1: The authors have addressed the few comments that I had, other than my point 1. I asked for a cartoon depicting the sleep system in flies, to help the non-expert reader along. Their answer is that "a cartoon that illustrates the multiple sleep-regulating regions in the fly brain would add confusion. ". This doesn't make any sense to me, why would adding a schematic cartoon add confusion. Anyhow, minor issue perhaps. I have no objections to the publication of this study at this stage.

However, I remain somewhat sceptical regarding the novelty of the findings and the appeal to the broad readership of PLoS Biology.

Reviewer #2: 

I find the authors' responses very clear, detailed and useful. I have these additional minor comments for the authors to consider. I think points 4/14 are the most important - discussion of the possibility that peripheral neurons/cells in the 23E10Gal4//23E10DBD lines (distinct from bowtie neurons) are sleep-promoting.

1. Line 67 Remarkably,....

 recommend you delete this word

2. Line 71 "... we sought …. " 

 The historical backdrop is not the obligatory basis for the narrative. I suggest the intro should finish much more strongly - emphasizing the significance of the effort/results.

3. Line 146 "... and is widely considered as a "dFB-specific" tool."

 This is an un-referenced observation - who says so and how widely-considered? 

4. Line 152. "Importantly, 23E10-GAL4 does not express in the legs, gut, or ovaries (Fig S1B-S1E) indicating that its sleep-promoting properties originate in the CNS." 

Based on that, the authors state: 'We found that 23E10-GAL4 does not express outside the CNS (Fig. S1B-S1E).'

 This conclusion is clearly premature; the survey of the periphery is largely incomplete. This is best done by staining sections of intact Drosophila. This is not a trivial point to my mind. The authors have argued effectively that the 23E10Gal4 line contains sleep-promoting cell types in addition to the dFSB ones. They showed that you must look outside the brain to find them and they found candidates in the ventral nerve cord. They were able to implicate 2 of those neurons by also finding them in different split gal4 lines which contained the same 23E10 DBD activity. I think the same logic that drove the authors to shine light on the bowtie neurons as candidate sleep regulators should prompt them to ask whether additional ones (or maybe even the "real" non-brain 23E10 sleep promotors) reside outside the CNS. If these exist, they may inextricably linked to the bowtie neurons in terms of enhancer activity, for the regulatory sequences tested. They should at least concede such neurons may exist. The finding of others (cited as #33) that leg cells have sleep-promoting activity serves as a clear precedent. (Also see point #14 below).

 Also, I recommend you delete the word "Importantly,"

5. Line 151 "….these results indicate that it is impossible to unequivocally assign a role in sleep promotion to 23E10-GAL4 dFB neurons (Fig 1I)."

 I think it fair at this point to re-visit the original Donlea et al experiments in which dFSB represented the overlap of three different positive Gal4 lines. Such independent application of different driver lines was not done here. Instead the 23E10 DBD line has properties equal to the original 23E10 Gal4.

6. Line 195: the 23E12-AD; 23E10-DBD Split-GAL4 "Surprisingly, our anatomical analyses revealed that it does not express in any dFB neurons (Fig 1O, Movie S3, Movie S4). Instead, this line expresses in two clusters of 4-5 neurons located in the anterior ventrolateral protocerebrum and in 4 VNC cells located in the metathoracic ganglion."

Then on line 211:

"Importantly, since all the neurons contained in the 23E12-AD; 23E10-DBD Split-GAL4 are part of the 23E10-GAL4 expression pattern, these data indicate that this driver contains non-dFB sleep-promoting cells."

 Question: are the two clusters of 4-5 neurons located in the anterior ventrolateral protocerebrum in the original 23E10-Gal4? I infer from the conclusion that they known to be sleep-promoting

7. Line 206 23E12-AD; 23E10-DBD additionally labels two neurons in the VNC whose somas are located very close to the "bowtie" neurons. These neurons have characteristic processes in each leg ganglion.

 How was it determined whose processes are whose, among the 4 VNC neurons?

8. Line 211 Importantly

 recommend you delete this word

9. Line 236 Remarkably

 recommend you delete this word

10. Line 354 We will from now on refer to these neurons as VNC-SP (VNC Sleep-Promoting). 

 Is that necessary? Why not stay with Bowtie neurons? There could be additional VNC-SP.

11. Line 459 Importantly

 recommend you delete this word

12. Line 460 the weak RNAi line does not reduce sleep significantly - 

 I recommend you make this point more strongly - it is very good for your argument.

13. Line 528 One of the goals of neuroscience is to ascribe behaviors to individual neurons. 

 Maybe refine this statement? Individual neurons are not absolute agents of control in all animals. Maybe 'ascribe to specific cell types'?

14. Line 539 Also unclear was whether the sleep promotion caused by activation of 23E10-GAL4 neurons is due to dFB neurons or to other cells within the 23E10-GAL4 pattern of expression.

 Again, I think this is over-stating the case against the original conclusions: three independent and different Gal4 lines were positive in that original screen - they shared the dFSB as a common point of expression in the brain. The present work has extended focus to the bowtie neurons of the VNC. But by the author's logic, it is fair to say it remains unclear whether the sleep promotion (caused by activation of the split Gal4 lines they have used) is due to BT neurons, or due to other (peripheral?) cells. (related to point #4 above).

15. Line 636 Note that a similar approach was used to validate ChAT RNAi constructs in other studies. 

 What other studies?

16. Fig 4 BC 

 There appear to be more than 1 BT neurons on each side. I do not see clear positive/negative signals for NT stains in the GABA panel.

Reviewer #3: The authors have addressed pretty much all the important issues raised by the reviewers and I think produced a manuscript ready for publication.

---

## [Editor Report · Decision Letter 3]

25 Jan 2023

Dear Dr Dissel,

Thank you for the submission of your revised Research Article "Regulation of sleep by cholinergic neurons located outside the central brain in Drosophila." for publication in PLOS Biology. On behalf of my colleagues and the Academic Editor, Maria Fernanda Ceriani, I am pleased to say that we can in principle accept your manuscript for publication, provided you address any remaining formatting and reporting issues. These will be detailed in an email you should receive within 2-3 business days from our colleagues in the journal operations team; no action is required from you until then. Please note that we will not be able to formally accept your manuscript and schedule it for publication until you have completed any requested changes. Also - one minor note, our Academic Editor thought that the new Fig1A resolution seemed a bit low. So you may want to check on this when submitting the final version of your work.

PRESS

We frequently collaborate with press offices. If your institution or institutions have a press office, please notify them about your upcoming paper at this point, to enable them to help maximize its impact. If the press office is planning to promote your findings, we would be grateful if they could coordinate with biologypress@plos.org. If you have previously opted in to the early version process, we ask that you notify us immediately of any press plans so that we may opt out on your behalf.

Sincerely, 

Kris

Kris Dickson, Ph.D., (she/her)

Neurosciences Senior Editor/Section Manager

PLOS Biology

kdickson@plos.org